# A remarkable genetic shift in a transmitted/founder virus broadens antibody responses against HIV-1

Swati Jain[1†], Gherman Uritskiy[1], Marthandan Mahalingam[1], Himanshu Batra[1‡, §], Subhash Chand[1#], Hung V Trinh[2,3], Charles Beck[4], Woong-Hee Shin[5,6,7¶], Wadad Alsalmi[1], Gustavo Kijak[2,3**], Leigh A Eller[2], Jerome Kim[3††], Daisuke Kihara[5,8], Sodsai Tovanabutra[2,3], Guido Ferrari[4], Merlin L Robb[2], Mangala Rao[3], Venigalla B Rao[1*]

[1]Bacteriophage Medical Research Center, Department of Biology, The Catholic University of America, Washington, United States; [2]Henry M. Jackson Foundation for the Advancement of Military Medicine, Bethesda, United States; [3]Laboratory of Adjuvant and Antigen Research, U.S. Military HIV Research Program, Walter Reed Army Institute of Research, Silver Spring, United States; [4]Department of Molecular Genetics and Microbiology, Duke University, Durham, United States; [5]Department of Biological Sciences, Purdue University, West Lafayette, United States; [6]Department of Chemistry Education, Sunchon National University, Suncheon, Republic of Korea; [7]Department of Advanced Components and Materials Engineering, Sunchon National University, Suncheon, Republic of Korea; [8]Department of Computer Science, Purdue University, West Lafayette, United States

*For correspondence: rao@cua.edu

Present address: †Department of Surgery (Head and Neck service), Memorial Sloan Kettering Cancer Center, New York, United States; ‡Howard Hughes Medical Institute, Program in Cellular and Molecular Medicine, Boston Children's Hospital, Boston, United States; §Department of Genetics, Harvard Medical School, Boston, United States; #Department of Anesthesiology, University of Nebraska Medical Center, Omaha, United States; ¶Department of Biomedical Informatics, Korea University College of Medicine, Seoul, South Korea; **Translational Medicine, Vaccines & Immune Therapies, BioPharmaceuticals R&D, AstraZeneca, Gaithersburg, United States; ††International Vaccine Institute, Seoul, South Korea

**Abstract** A productive HIV-1 infection in humans is often established by transmission and propagation of a single transmitted/founder (T/F) virus, which then evolves into a complex mixture of variants during the lifetime of infection. An effective HIV-1 vaccine should elicit broad immune responses in order to block the entry of diverse T/F viruses. Currently, no such vaccine exists. An in-depth study of escape variants emerging under host immune pressure during very early stages of infection might provide insights into such a HIV-1 vaccine design. Here, in a rare longitudinal study involving HIV-1 infected individuals just days after infection in the absence of antiretroviral therapy, we discovered a remarkable genetic shift that resulted in near complete disappearance of the original T/F virus and appearance of a variant with H173Y mutation in the variable V2 domain of the HIV-1 envelope protein. This coincided with the disappearance of the first wave of strictly H173-specific antibodies and emergence of a second wave of Y173-specific antibodies with increased breadth. Structural analyses indicated conformational dynamism of the envelope protein which likely allowed selection of escape variants with a conformational switch in the V2 domain from an α-helix (H173) to a β-strand (Y173) and induction of broadly reactive antibody responses. This differential breadth due to a single mutational change was also recapitulated in a mouse model. Rationally designed combinatorial libraries containing 54 conformational variants of V2 domain around position 173 further demonstrated increased breadth of antibody responses elicited to diverse HIV-1 envelope proteins. These results offer new insights into designing broadly effective HIV-1 vaccines.

## eLife assessment

This study provides a detailed evaluation of how HIV evades nascent immune pressure from people living with HIV followed nearly immediately after infection. There is **convincing** evidence that H173

mutations in the V2 loop was a key determinant of selection pressure and escape. These data are congruent with protection in the RV144 clinical trial, the only trial that showed protection from infection. Overall, this study is an **important** contribution to the field.

## Introduction

While combinatorial antiretroviral therapy has greatly improved the life expectancy of people living with HIV, it does not cure the infection even with life-long adherence (*Mocroft et al., 2007*; *WHO, 2021*). In the absence of a preventative vaccine, HIV-1 continues to be a global public health concern, causing ~1.5 million new infections annually (*Global HIV & AIDS statistics, 2019*). After dozens of HIV-1 vaccine failures in the last four decades, the only vaccine trial that showed promise was the phase 3 RV144 trial conducted in Thailand (*Esparza, 2013*; *Gilbert et al., 2011*; *Rerks-Ngarm et al., 2009*; *de Souza et al., 2012*; *Haynes et al., 2012*). The RV144 demonstrated an early efficacy of ~60% reduction in HIV acquisition at 12 months post-vaccination which gradually declined to 31.2% at 42 months (*Gilbert et al., 2011*; *Rerks-Ngarm et al., 2009*). Several studies demonstrated correlation of IgG antibodies specific to the V2 variable domain of the HIV envelope protein (Env) to vaccine efficacy (*de Souza et al., 2012*; *Haynes et al., 2012*; *Zolla-Pazner et al., 2013*; *Zolla-Pazner et al., 2019*). Notably, protection was not due to their ability to neutralize the virus but likely due to the Fc effector function, specifically the antibody-dependent cell cytotoxicity (ADCC) (*Haynes et al., 2012*). Furthermore, sieve analysis of the breakthrough infections in RV144 vaccinees showed mutations in the V2 domain, in the semi-conserved structural core encompassing residues 166–183, which seems to be one of the prime targets of the host immune pressure (*Rolland et al., 2012*; *Moodie et al., 2022*). Therefore, broadening the vaccine-induced immunity against the V2 region can potentially confer protection against breakthrough infections. Additionally, there have been concerted efforts to generate vaccine-induced broadly neutralizing antibodies (*Jardine et al., 2016*; *Leggat et al., 2022*; *Lee et al., 2017*; *Jardine et al., 2015*) but remains challenging and elusive.

The Env spike on the surface of HIV forms essential interactions with its primary receptor, CD4, followed by interactions with the CCR5/CXCR4 co-receptor on a CD4 +T cell for viral entry (*Kwong et al., 1998*; *Pancera et al., 2014*; *Klasse, 2012*; *Wilen et al., 2012*). In addition to CD4 and CCR5/CXCR4 receptors, interaction of the V2 region with integrin α4β7 has also been implicated as a significant contributor in the pathogenesis of HIV-1, particularly for dissemination and gut-reservoir establishment in the infected individuals (*Arthos et al., 2008*; *Arthos et al., 2018*). It has been shown that the V2 domain by virtue of mimicking MadCAM, a natural ligand of α4β7, assists in co-stimulation of CD4 +T cells promoting HIV-1 replication during an acute stage of infection (*Goes et al., 2020*). Furthermore, vaccinees of the RV144 trial generated non-neutralizing V2 antibodies that are shown to block interaction with α4β7 (*Peachman et al., 2015*; *van Eeden et al., 2018*; *Lertjuthaporn et al., 2018*).

Env is expressed as a 160 kD glycoprotein (gp160) and cleaved by the cellular protease furin into gp120 and gp41 subunits. The membrane-external subunit, gp120, has five conserved regions (C1, C2, C3, C4, and C5) and five variable regions (V1, V2, V3, V4, and V5) that are alternately positioned in the Env sequence with the exception of V1V2 variable regions that assemble as a single domain (*Land and Braakman, 2001*; *Leonard et al., 1990*). Three protomers, each composed of non-covalently associated gp120 and gp41 subunits, assemble as a trimeric spike on the viral envelope. The three V1V2 domains form a well-exposed 'crown' of the mushroom-shaped spike, hence a frequent target of the host immune system. Each V1V2 domain consists of a conserved Greek-key motif structure with 4–5 β-strands (A, B, C, C', D) forming an anti-parallel β-sheet and two hypervariable loops that are flexible and conformationally dynamic (*Pan et al., 2015*). The virus takes advantage of these features to engage in immune battles with the host defenses and selecting mutations, particularly in the hypervariable loops that can escape host immunity (*Trinh et al., 2019*; *Hioe et al., 2018*; *Rao et al., 2013*; *Sagar et al., 2006*). The V1V2 domain of HIV-1 envelope protein is therefore an attractive target for a preventative vaccine design.

The human host is often exposed to a complex genetic pool of highly diverse and heterogeneous viral quasispecies from an infected donor of which only one (or a few) transmitted founder (T/F) virus can successfully establish a productive infection (*Keele et al., 2008*). However, very little is known about how the T/F viruses emerge out of the exposed variants or how they might escape

the initial host immune responses to establish chronic infections. Moreover, how the structural and conformational dynamics of the variable V1V2 domain might contribute to mechanisms of immune escape remains poorly understood. Here, we investigated the mechanism of escape of a T/F virus from V2-directed host immune responses in an acutely infected participant of a prospective acute HIV cohort, RV217 (*Robb et al., 2016*). RV217 is a unique and rare clinical trial in which HIV infection could be tracked starting from just days after infection up to nearly 3 years in the absence of antiretroviral therapy (*Robb et al., 2016*; *Ananworanich et al., 2017*). Our studies discovered a remarkable genetic shift in the T/F virus population due to a key mutation, H173Y, in the semi-conserved epitope of the V2 domain, which rendered the pre-existing antibody responses completely obsolete. Furthermore, this mutation caused a conformational shift of the V2-epitope to which the human host responded with more broadly reactive antibody responses, a phenomenon that we show could be recapitulated in a mouse model. We further show that combinatorial V2 immunogen libraries comprised of 54 naturally occurring conformational variants at or near H173 elicited antibody responses with increased breadth to diverse HIV-1 envelope proteins. These studies provide new insights into effective design of future HIV vaccines that could induce broad immunity against diverse strains of HIV-1.

## Results
### Hypothesis and experimental design
We hypothesized that understanding how a T/F virus escapes host immune pressure at the very early stages of infection might identify key mutations that if included in a HIV vaccine could stimulate broader immune responses and preemptively interfere or block HIV acquisition. Our primary focus

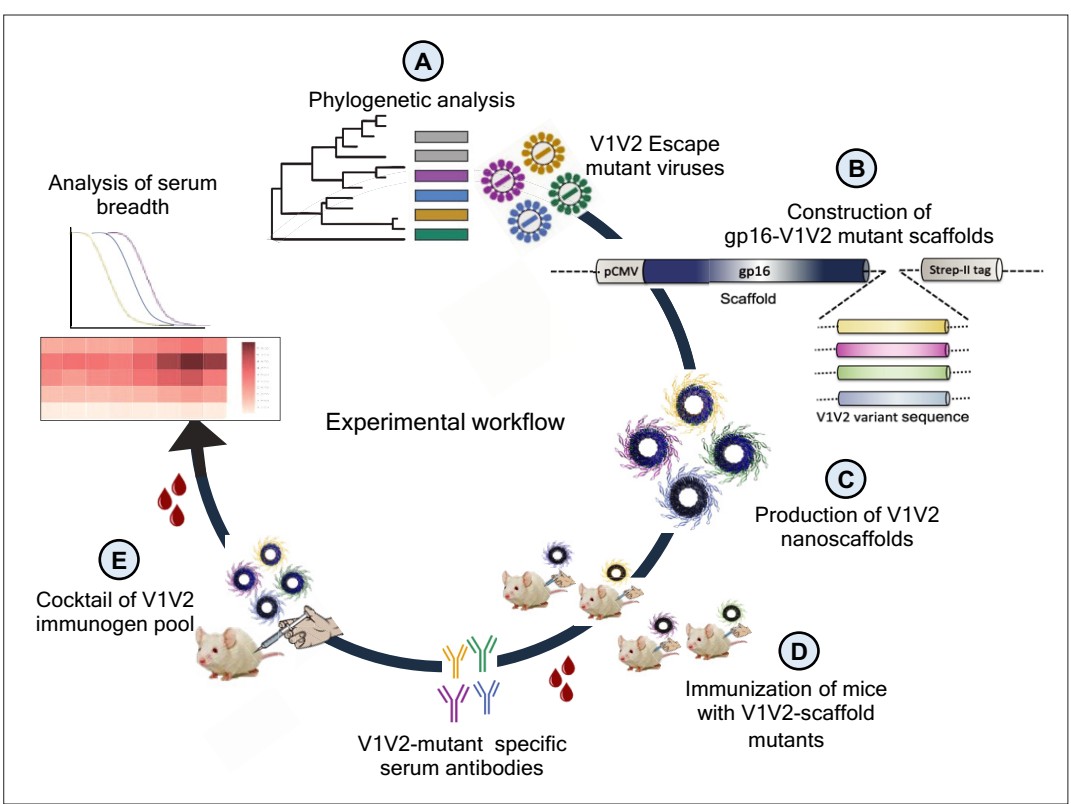

**Figure 1.** Experimental design. (**A**) Phylogenetic analysis of longitudinal *env* sequences isolated from four acutely infected participants in the RV217 study. (**B**) Selection of V1V2-specific escape variants and their fusion to the C-terminus of bacteriophage T4 small terminase protein, gp16, to generate gp16-V1V2 nanoscaffolds representing various escape mutations. (**C**) Expression of gp16-V1V2 escape mutants in GnTi mammalian cells to produce dodecameric nanoscaffolds displaying variant immunogens. (**D**) Immunization of mice with V1V2 variant nanoscaffolds and evaluation of immunogenicity. (**E**) Design of combinatorial V1V2 mutant immunogens and evaluation of their immunogenicity and breadth of immune responses in a successive immunization study.

is the V1V2 domain of HIV-1 Env, one of the key determinants of virus escape in the moderately successful RV144 vaccine trial (*Rolland et al., 2012*; *Liao et al., 2013*). We therefore decided to study the evolution of V1V2 domain in the T/F viruses from four HIV-1 infected participants of RV217 Early Capture HIV Cohort (ECHO) project, a unique and rare trial that allowed tracking of HIV infection starting from just days after infection in the absence of antiretroviral therapy.

First, we performed phylogenetic analyses of longitudinal Env sequences to identify V1V2-specific variants that correlated with T/F virus escape (*Figure 1A*). Next, the selected escape variants were fused with the oligomeric (11- and 12-mers) bacteriophage (phage) T4 small 'terminase' protein, gp16, to generate V1V2 epitope-displaying nanoscaffolds (*Sun et al., 2012*) for effective antigen presentation (*Figure 1B*). These constructs were then expressed in HEK293GnTi (GnTi) cells to produce gp16-scaffolded V1V2 escape mutant domains with native-like high mannose glycosylation (*Figure 1C*). Such glycosylated gp16-V1V2 scaffolds were previously shown to retain a native-like V1V2 epitope presentation as evident from their binding to well-characterized mAbs, in particular the glycan-dependent PG9 antibody (*Chand et al., 2017*). Next, the immunogenicity of the V1V2 variants was evaluated in a mouse model to determine if they could recapitulate the distinctive escape mutant-specific immune responses observed in the human plasma (*Figure 1D*). Finally, cocktails of V1V2 variant-targeted immunogen pools were created using controlled mutagenesis to determine the effect of escape mutation on the antibody response and recognition of diverse HIV-1 Env antigens in the context of a library designed with and without escape mutations (*Figure 1E*). Since V2 is known to be a very conformationally polymorphic region of the Env, we hypothesized that if the escape mutations identified in the T/F virus would have dominant effect of on the conformation and subsequent antibody response despite other V2 mutations. Identifying such conformation influencing V2 residues might inform effective vaccine design.

## Phylogenetic analyses identified a striking H173Y escape mutation in the V2 domain of a T/F virus

A series of HIV-1 viral *env* sequences were isolated from four HIV-1 infected participants (*referred here as participants – 7, 61, 94 and 100*) of the RV217 ECHO project through single genome amplification (SGA). Using a sensitive nucleic acid test, each study participant was confirmed of HIV-1 positivity just days after a negative test (Fiebig stage I) and none of these participants were on antiretroviral therapy during the timeframe of the study. *Env* sequences were obtained at three time points, wks 1, 4 (~1 month), and 24 (~6 months) following the positive test. That a single T/F virus was responsible for infection in each participant was further ascertained by aligning independently isolated *env* sequences from wk 1 plasma, which were nearly identical (*Robb et al., 2016*). Additionally, plasma viral load with peak, nadir, and set point viremia shown for one representative patient (participant 7) indicated a typical pattern of early captured infection (*Figure 2—figure supplement 1*). Around 30 sequences were analyzed from each patient (median of 10 sequences per visit), with 152 sequences in total across all four patients, over a period of up to 6 months post-infection.

To trace viral phylogeny for each of the T/F viruses (hereafter referred to as T/F07, T/F61, T/F94, and T/F100), the longitudinal *env* sequences were translated to protein sequences and aligned. Multiple sequence alignments were then used to construct phylogenetic trees using the respective T/F virus *env* sequence as the root for the tree construction (*Figure 2—figure supplement 2*). Of the four T/F viruses, T/F61 showed a few dominant mutations with relatively conserved V1V2 region, while T/F100 was found to be the most rapidly diverged virus with mutants appearing at as early as one wk after infection. By wk 4, a major branch of diverging T/F100 viruses harboring mutations in the V1V2 domain appeared. In addition, there were mutations in V5 variable loop and the less conserved α2 helix of C3 constant region. T/F94 virus also acquired various mutations including deletions in the variable V1V2 and V5 regions. Not surprisingly, most of these mutations are in the surface-exposed variable regions of HIV-1 trimer with hotspots in the loop regions (*Figure 2A–B*).

On the other hand, T/F07 virus did not show any variants until wk 4. At 24 wk post-infection, however, nearly the entire T/F07 virus population made a complete shift to a single variant containing two mutations, one at position 173 that changed histidine to tyrosine (H173Y) and another at position 236 that changed lysine to threonine (K236T). The K236T mutation restored the well-conserved N-linked glycan at position N234 located at gp120/gp41 interface. Additionally, in some of the variants, a 3-residue deletion in the variable V2 loop (ΔDSV) and a 5-residue deletion (ΔNTTRFL) in the

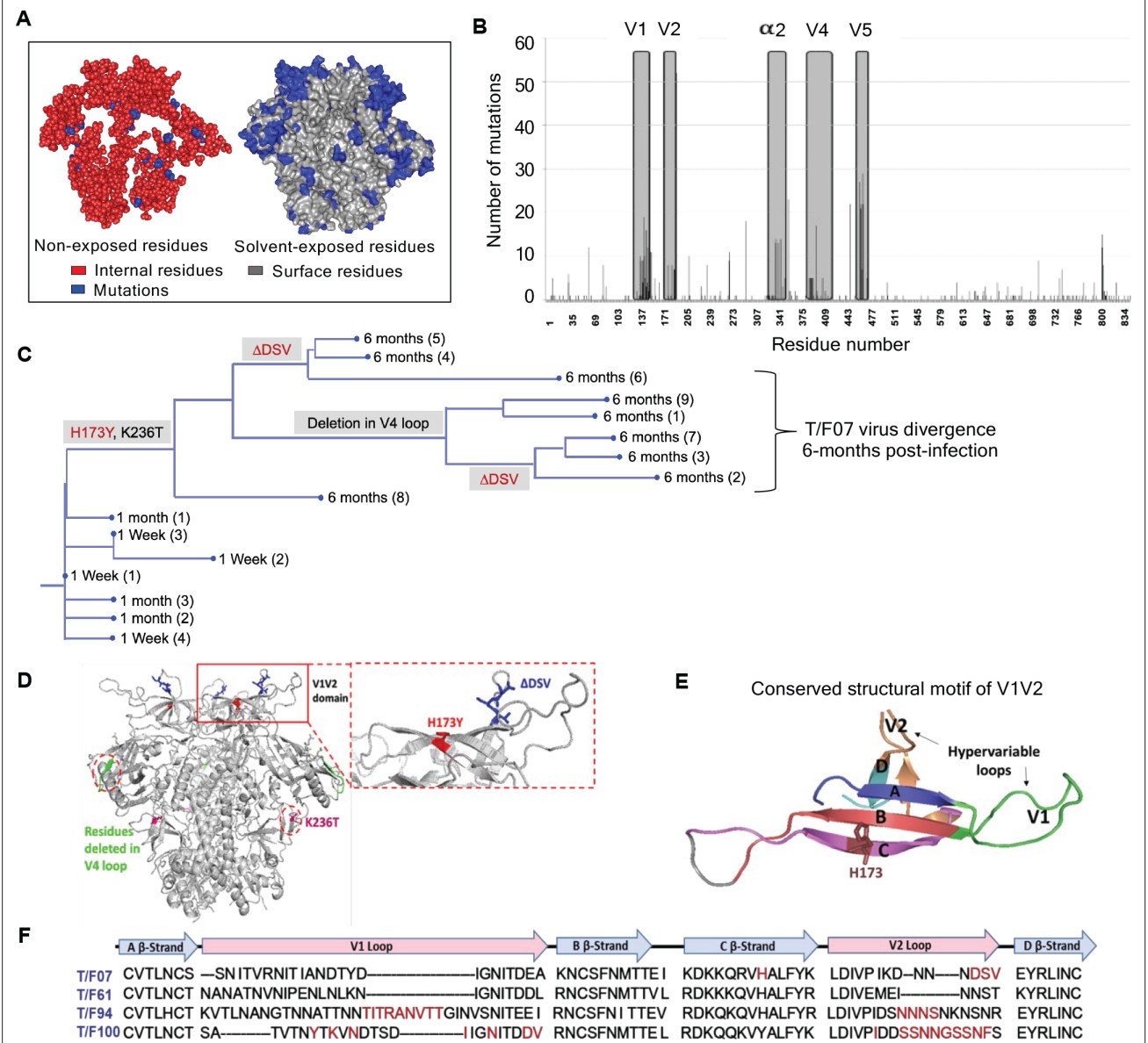

**Figure 2.** Viral escape mutations in RV217 participants during the acute stage of HIV infection. (**A**) Mutations observed in the RV217 participants – 7, 61, 94, and 100 are placed on HIV-1 BG505 pre-fusion trimer structure (PDB ID: 4TVP). Surface model showing dominant mutations (blue) falling on the solvent-exposed regions of the Env (gray) but not on the non-exposed or buried region (red). Modeling was done with PyMol (ver. 1.74) molecular visualization software and surface-exposed residues were defined as all residues that had >5 Å² exposure to the solvent. (**B**) Mutational hotspots in the *env* sequence based on the genetic diversification of T/F viruses from all four participants. Total number of mutations in a particular region is plotted on the y-axis against residue positions on the *env* sequence (x-axis), with reference to HXB2 strain. The Env regions (labeled on top) with gray background showed high frequency of mutations. (**C**) Phylogenetic tree displaying T/F07 virus evolution in participant_7. The evolutionary tree was constructed by the neighbor-joining method, rooted to the T/F07 virus sequence. The viruses are designated corresponding to the time post-infection; 1 wk, 1 month (4 wks) or 6 month (24 wks) at the nodes of the branches. Prominent diverging mutations are labeled on the respective branches of the tree in red (V1V2 region) and gray (another region). (**D**) Dominant mutations that occurred until 24 wk post-infection in participant_7 are modeled on a ribbon model of the T/F07 trimer, generated through homology modeling using BG505 trimer (PDB ID: 4TVP) as a template. The zoomed-in image of V1V2 domain is shown to highlight the positions of V2-specific mutations, H173Y (red) and 3-residue deletion, DSV (blue). Deletion in the variable, V4 region is depicted in bright green and mutation in the conserved C2 region, K236T substitution, is shown in magenta. (**E**) A color-coded 4–5 β-stranded (**A–D**) conserved Greek-key motif structure of V1V2 domain is represented showing residue 173 on the C β-strand. (**F**) Snapshot of V1V2 sequences of T/F viruses from each RV217 participant under study. Major structural features of V1V2 region; semi-conserved four β-strands (**A, B, C and D**) and hypervariable V1 and V2 loops are labeled on top of the sequence. Prominent V1V2-specific mutation sites (observed in >50% of circulating viruses) until 24 wks are highlighted in red in the respective T/F virus sequence isolated from participants – 7, 61, 94, and 100.

*Figure 2 continued on next page*

*Figure 2 continued*

The online version of this article includes the following source data and figure supplement(s) for figure 2:

**Figure supplement 1.** Longitudinal viral load analysis over a course of infection in the participant 7.

**Figure supplement 1—source data 1.** Data used for generating graph in *Figure 2—figure supplement 1*.

**Figure supplement 2.** Phylogenetic trees showing evolution of transmitted/founder viruses.

variable V4 loop co-occurred with the H173Y mutation (*Figure 2C–D*). The H173Y mutation is localized in the relatively well-conserved 'C' β-strand positioned at the junction of the cationic first half and the hydrophobic second half of the β-strand (*Figure 2E*). These characteristics, a singular variant and the remarkable shift of the viral population, strongly suggested a linkage between the variant and potential viral escape. This was in contrast to many mutations observed in the other three T/F viruses in the hypervariable V1 and V2 loops (*Figure 2F*) which are difficult to track and of little value for vaccine design. Furthermore, the distinct (and complex) evolutionary trajectories taken by different T/F viruses highlighted the breadth of immune responses needed for an effective HIV-1 vaccine design.

## The H173Y mutation in C β-strand of V2 domain is a key determinant of virus escape against host immune pressure

The divergence of nearly the entire T/F07 virus population to H173Y variant made it a strong candidate for a viral escape mechanism. H173 is located in the C β-strand of V1V2 domain, a region that in previous studies was also found to be a critical target for host immune responses by RV144 trial vaccines. To determine if this mutant indeed arose through a strong selection against V2-directed antibody responses, the epitope specificity of antibodies was evaluated in the longitudinal plasma samples. The V1V2 domains, but not the full-length Env proteins, were used for testing in order to exclude the binding responses directed against other regions of the envelope protein. Four V1V2 domain recombinants were constructed using the phage T4 small terminase subunit gp16 (18 kDa) as a nanoscaffold: gp16-H173 ('wild-type' T/F07) and three 24-wk V2 mutants namely, gp16-H173Y (Y173), gp16-ΔDSV, and gp16-Y173+ΔDSV. Gp16 is a well-characterized oligomer that forms highly soluble and stable 11-mer and 12-mer ring structures. The V1V2 domains with a StrepTag fused to the C-terminus of gp16 are expected to decorate the nanoscaffold. These were expressed in GnTi cells, affinity-purified by StrepTactin chromatography (*Figure 3—figure supplement 1*) and used to capture the respective V1V2-specific antibody titers in the longitudinal plasma samples of participant 7 by surface plasmon resonance (SPR) assay.

The data revealed remarkable epitope specificity to the V2 domain of the original T/F07 virus, but not to the H173Y variant, at 24 week post-infection. The antibodies bound strongly to V1V2-H173 and V1V2-H173.ΔDSV scaffolds but failed to bind to the Y173 variant scaffolds V1V2-Y173 and V1V2-Y173.ΔDSV (*Figure 3A*). These data demonstrated that the H173Y mutation is a key determinant in epitope switching at the time of virus escape against host immune pressure. No significant difference was observed with the DSV deletion mutants, although it might have played an accessory role (see below).

Following this first 'wave' of antibodies specific to T/F07-H173 epitope, there was a second wave after the virus escape when the virus population switched to the resistant Y173 variant. SPR analyses showed that, contrary to the first wave, this second wave of antibodies exhibited increased breadth, recognizing both the V1V2-H173 and V1V2-Y173 variants (*Figure 3A*; see the peak at 1.4 years). Therefore, the second wave antibodies would be able to restrict both the original T/F07 H173 virus as well as the escaped Y173 variant. Collectively, these data strongly suggest that the V2-region, in particular the C β-strand, is a critical target for mounting immune pressure by host during the acute HIV infection.

## Recapitulation of escape mutant specificity in monoclonal antibodies from RV144 vaccinees

Next, we evaluated the binding of V2-H173 and V2-Y173 variants to the C β-strand-specific monoclonal antibodies (mAbs) CH58 and CH59 that were isolated from the RV144 trial vaccinees. These antibodies were previously reported to bind to H173 C β-strand present in the gp120 immunogen used in the RV144 vaccine trial, and the moderate protection observed in the RV144 trial correlated with such V2 C β-strand-specific antibody responses. Remarkably, while the V1V2-H173 scaffolds

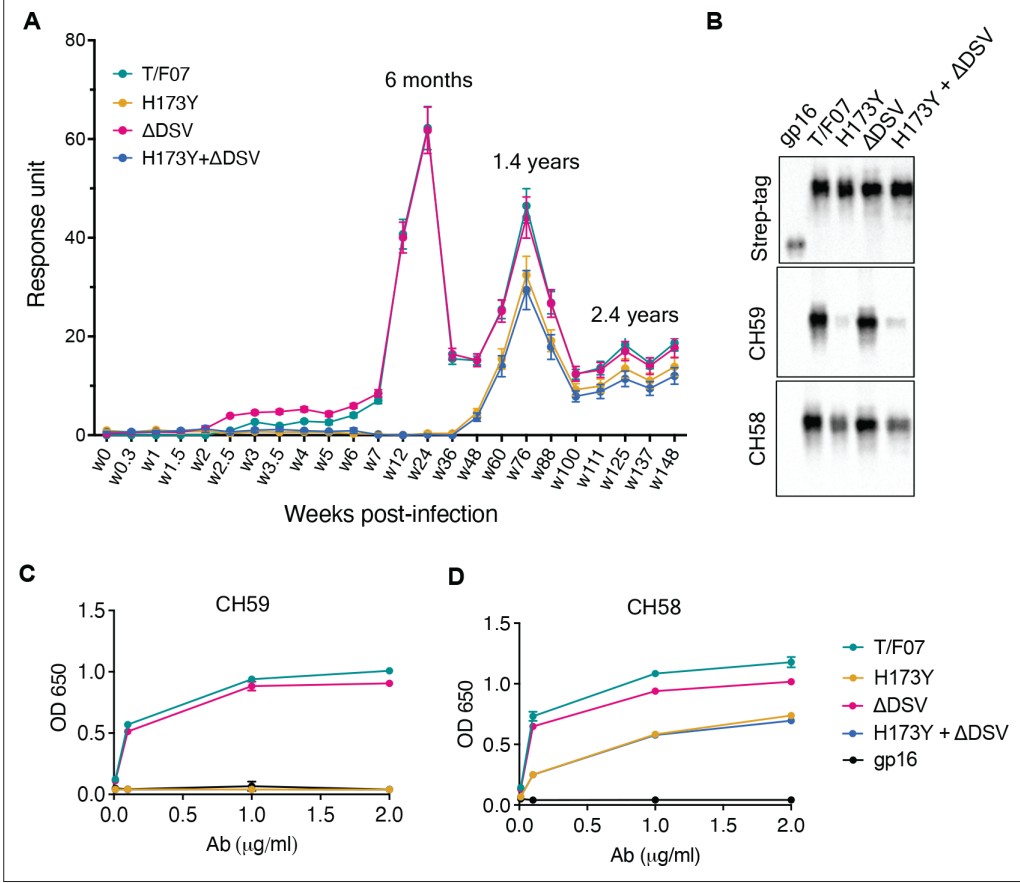

**Figure 3.** A H173Y mutation in T/F07 HIV-1 Env led to virus escape in the participant 7. (**A**) SPR binding curves showing reactivity of purified recombinant gp16-T/F07 (green), gp16-H173Y (red), gp16-ΔDSV (magenta) and gp16-H173Y+ΔDSV (blue) V1V2 proteins with a series of longitudinal plasma samples collected from participant 7 (shown on x-axis). The samples from wk 0 (pre-infection) until wk 144 (post-infection) were tested. The amount of binding is proportional to the response units (RU) plotted on the y-axis. The corresponding time points for each peak of antibodies are indicated. An early wave of V2-specific antibody represented by first peak at 6-month time-point recognized the gp16-T/F07 and gp16-ΔDSV variants [H173 variants] but not the gp16-H173Y and gp16-H173Y+ΔDSV variants [Y173 variants]. The second and third peaks of antibodies were found to be reactive to both the variants. (**B–D**) H173Y mutants poorly react to CH59 and CH58 antibodies. Immunoblot of the purified gp16 T/F07 V1V2 mutants (labeled on the top) showing respective binding with V2 mAbs, CH59 and CH58. Recognition by Strep-tag (purification tag) antibody served as a protein loading control (**B**). Binding curves of gp16-T/F07 (green), gp16-H173Y (red), gp16-ΔDSV (magenta), gp16-H173Y+ΔDSV (blue) V1V2-proteins, and gp16 scaffold only (negative control) (black) showing reactivity to CH59 (**C**) and CH58 (**D**) mAbs, as determined by ELISA performed with three technical replicates.

The online version of this article includes the following source data and figure supplement(s) for figure 3:

**Source data 1.** Uncropped Western blot images.

**Source data 2.** Data used for generating graphs in *Figure 3A and C-D*.

**Figure supplement 1.** gp16-V1V2 construct design and purification.

bound strongly to both CH58 and CH59 antibodies, the V1V2-Y173 escape variant either failed to bind (CH59) or showed drastically reduced binding (CH58). The DSV deletion again did not show a significant difference in the binding specificity (*Figure 3B–D*).

We then evaluated the ADCC responses as these were identified as one of the correlates for protection in RV144 vaccinees (*Mayr et al., 2017*; *Bonsignori et al., 2012*; *Bradley et al., 2017*; *Pollara et al., 2014*). We determined the relative ADCC responses in the participant 7 plasma against the C β-strand variants. We constructed the gp120 ectodomain versions of the V1V2 domain variants; gp120-H173 (T/F07), gp120-Y173, and gp120-Y173.ΔDSV (escape mutants), and gp120-92Th023

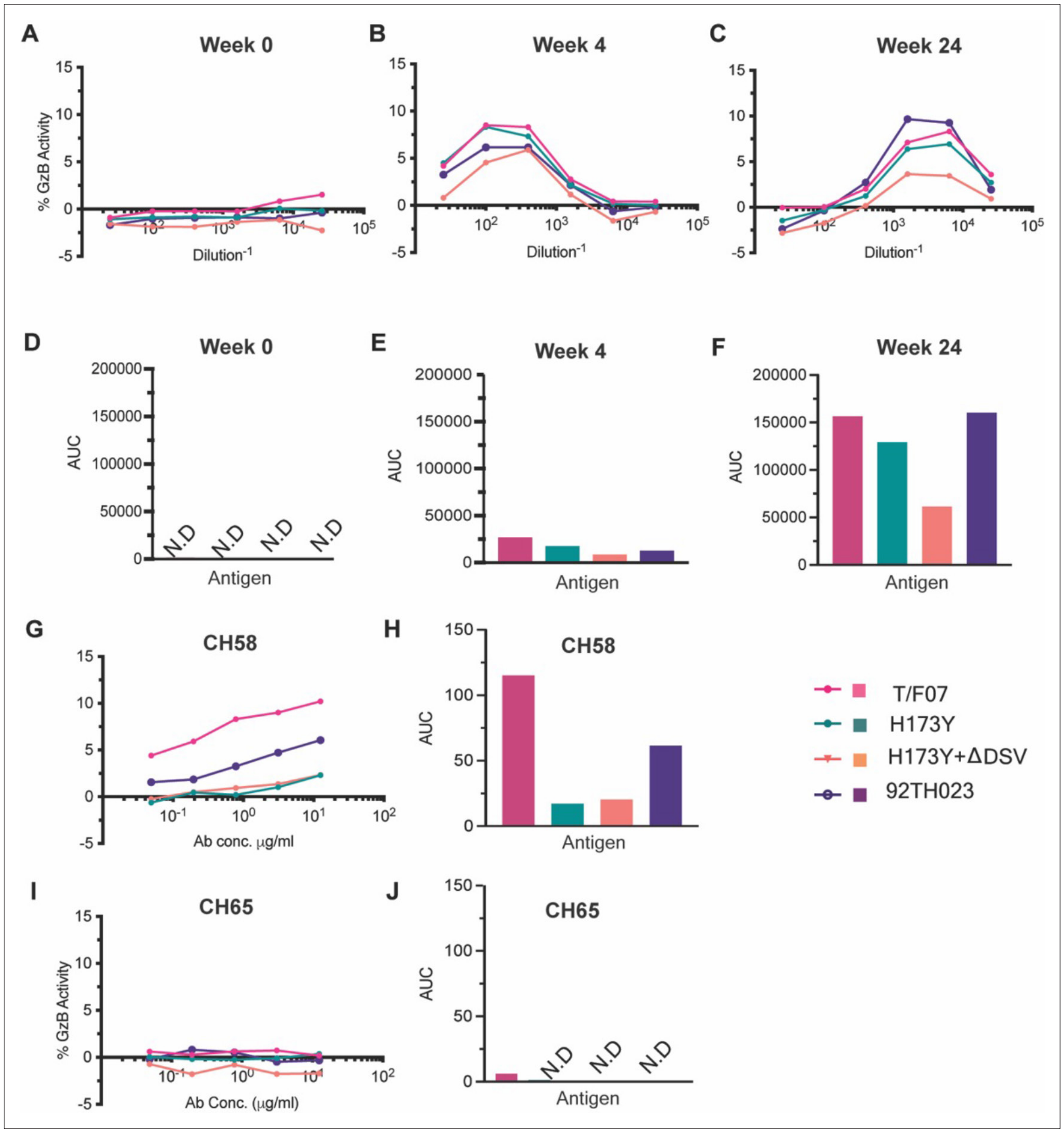


**Figure 4.** ADCC responses in longitudinal plasma samples of participant 7. (**A–C**) ADCC responses at wk 0 (A, pre-infection), 4 wks post-infection (**B**), 24 wks post-infection (**C**) of participant 7 plasma samples measured as percentage Granzyme activity (y-axis), against gp120-T/F07 (pink), gp120-H173Y (teal), gp120-H173Y+ΔDSV (orange) and control antigen, gp120-92TH023 (violet) coated target cells. (**D–F**) The corresponding area under curve (AUC) values for A-C graphs are shown for each curve. (**G–J**) V2-specific CH58 positive control antibody, (**G–H**) and Flu-specific CH65 negative control antibody, (**I–J**) mediated ADCC responses are shown with corresponding AUC values. For CH65 and CH58 mAbs, 4-fold serial dilutions starting at 12.5 μg/ml concentration were used as shown on the x-axis. Participant 7 plasma samples from each visit were 5-fold serially diluted to determine the ADCC responses. N.D is not detected.

The online version of this article includes the following source data for figure 4:

**Source data 1.** Data used to generate graphs in ***Figure 4***.

(control), expressed each in GnTi cells, and purified the recombinant proteins. These were then coated on the target cells and ADCC killing by effector cells (healthy PBMCs) was measured by Granzyme activity using wk0, wk4 and wk24 plasma samples of participant 7. The data showed moderate ADCC responses against the H173 T/F07, whereas resistance to killing was observed for the Y173 mutants, with the Y173.ΔDSV double mutant showing more resistance than the Y173 single mutant (***Figure 4***). This implicates a potential functional role for ΔDSV mutation against ADCC-type immune pressure, although it had no significant impact on recognizing the binding epitope(s). Furthermore, significant

resistance was also observed for both the Y173 variants against CH58 mAb-mediated ADCC killing, while no resistance was detected against a negative control CH65 Flu antibody.

The above data strongly implicated the H173Y mutation as the key variant selected against the host's immune pressure directed against the V2-domain of the original T/F07 HIV-1 virus that severely restricted virus survival.

## Recapitulation of V2-specific human immune responses in mice

We then hypothesized that the dramatic escape of H173Y mutant viruses might be because the histidine to tyrosine substitution caused a significant structural/conformational change in the C β-strand epitope such that it is no longer recognized by H173-specific antibodies. There is evidence that the C β-strand is conformationally dynamic and that it can take a helical form when bound to certain antibodies (*Liao et al., 2013*; *Wibmer et al., 2018*). This is also consistent with the distinct specificities of human antibodies generated against these variants, in participant 7, i.e., strict specificity of T/F07

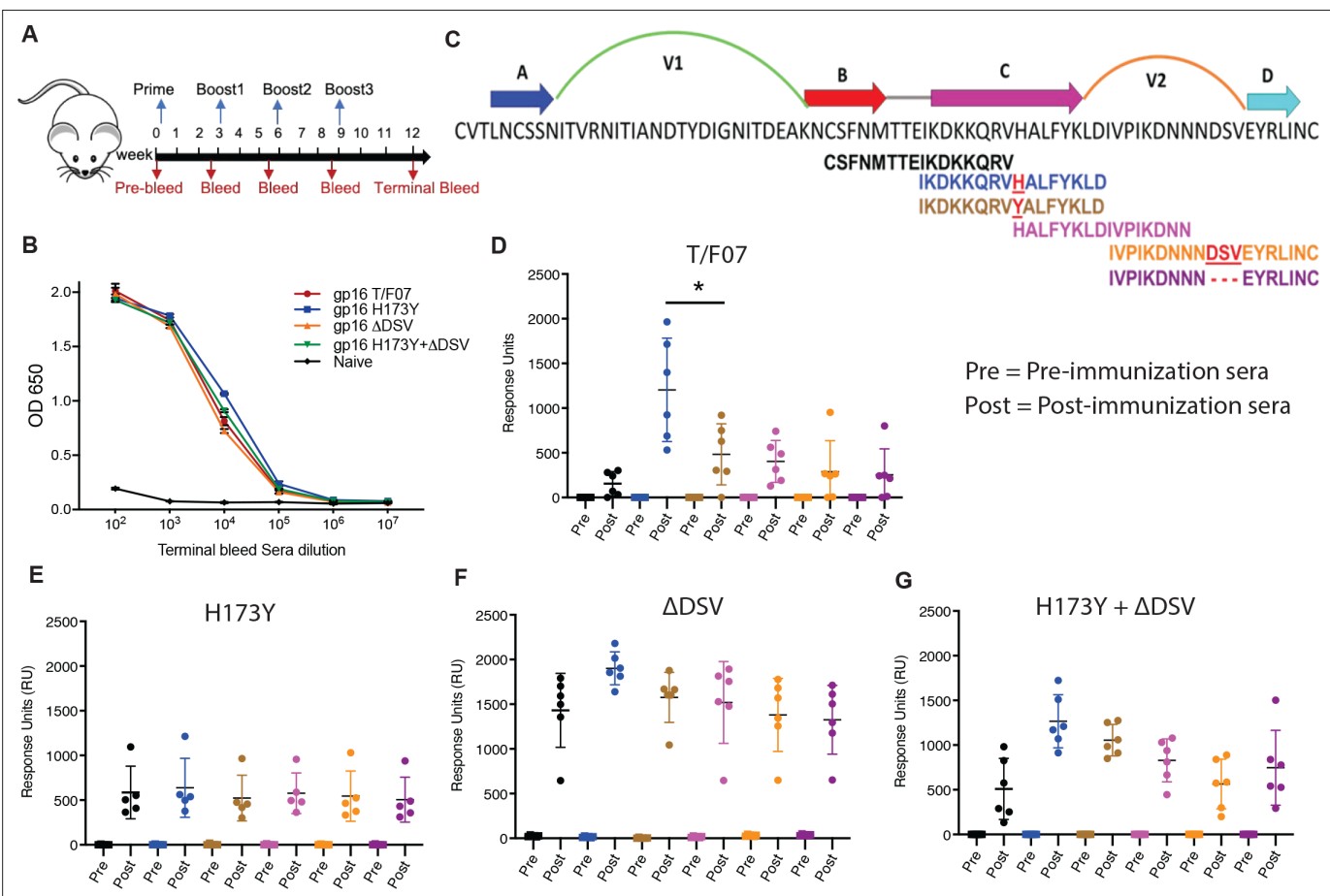

**Figure 5.** Binding patterns of V1V2 variant-immunized mice sera with overlapping V2-peptides. (**A**) Immunization scheme. Prime plus 3-boost immunizations were performed at an interval of three wks. Pre-bleed sera were collected prior to immunizations. (**B**) V1V2-specific responses in different immunized groups after the last immunization (terminal bleed). Binding curves for each group are color-coded, as shown on the right side of the graph. (**C**) Schematic showing T/F07-V1V2 sequence labeled on top for A-D β-strands and hypervariable V1 and V2 loops. The overlapping V2 peptides used for SPR binding analysis with immunized mice sera are shown in different colors. (**D–G**) SPR binding responses shown for gp16-T/F07 (**D**), gp16-H173Y (**E**), gp16-ΔDSV (**F**), gp16-H173Y+ΔDSV (**G**) groups determined 2 weeks post-immunization (post) of the last boost with specific V2 peptides represented by the corresponding color of the peptides shown in (**C**). Pre-bleed sera prior to immunization sera were used as negative controls. The assay was performed with four technical replicates. Each dot represents individual mouse sera. The binding signal is estimated as Response Units (RU) shown on the y-axis. p-Value was determined through unpaired t-test, p-value = 0.0252.

The online version of this article includes the following source data and figure supplement(s) for figure 5:

**Source data 1.** Data used to generate graphs in *Figure 5D-G*.

**Figure supplement 1.** V1V2-specific antibody titers.

plasma for H173 and increased breadth in the case of escaped plasma for both H173 and Y173 (as shown in *Figure 3*). Could this be recapitulated through immunogen design, in the absence of viral infection? To address this question, we immunized BALB/c mice (6 mice/group) with V1V2 nanoscaffolds containing four different V2 variants; H173, Y173, ΔDSV, and Y173.ΔDSV, and analyzed the specificity of the elicited antibody responses (*Figure 5A*).

To determine the V2 variant epitope-specific antibody titers, we constructed the His-tagged gp140 Env recombinants H173, Y173, ΔDSV, or Y173.ΔDS, expressed in GnTi cell, and the purified Env proteins were used as the coating antigens. These gp140 Envs allowed evaluation of V2 epitope specificity in the native gp140 Env context and also filtered out the gp16 scaffold- and strep-tag-specific antibody titers. The ELISA data showed that all the V1V2 variant scaffolds elicited robust V2-specific antibodies that recognized the V2 epitopes in the context of gp140 Env (*Figure 5—figure supplement 1* and *Figure 5B*). This is notable because the V2 antibodies elicited against multiple scaffolded V2 immunogens tested previously, did not recognize the V2 epitopes in gp140 Env context, though they bound to the scaffold used for immunization (*Hessell et al., 2019*). The end point titers were on the order of ~$10^5$ in the immunized mice while the naïve controls showed no significant V2-specific titers. These results demonstrated high level of immunogenicity and specificity to native Env by the gp16-scaffolded V1V2 variants.

We then tested the epitope specificity of the V2-specific antibodies using a series of 15-mer biotinylated peptides spanning the C β-strand by a sensitive Biacore SPR assay (*Figure 5C*). Remarkably, the T/F07 H173-induced sera reacted strongly with H173 C strand peptide but poorly with Y173 and other variant peptides (*Figure 5D*). In contrast, the escape mutant Y173-induced sera showed broad reactivity to both H173 and Y173 peptides as well as to other variant peptides containing C β-strand epitope. Moreover, the level of reactivity of Y173 antibodies to peptides was low overall when compared to the same with the gp140 Env proteins probably because these antibodies are conformation-specific (*Figure 5E*). Thus, the mice sera recapitulated the behavior of H173 and Y173 antibodies produced in a human infection. Furthermore, the ΔDSV deletion enhanced and broadened the reactivity of the antibodies (*Figure 5F–G*), which is consistent with the ADCC assays where the ΔDSV mutation showed enhanced resistance to cell killing.

## H173 and Y173 variants induce antibodies of distinct specificities

To further define the specificities of H173- and Y173-induced sera, we evaluated their ability to recognize HIV-1 Env proteins from different clades and determine their cross-reactivity. Accordingly, we constructed a series of recombinant clones and purified gp140 Env proteins from different T/F viruses, mutants, and clades: CRF_AE proteins T/F07-H173, T/F07-H173Y, T/F07-ΔDSV, T/F07-H173Y.ΔDSV, T/F61, T/F94, and T/F100, clade A [BG505], clade B [SF162 and JRFL], and clade C [1086] (*Figure 6A*). Their reactivity was tested using sera of mice immunized with the four V1V2 nanoscaffolds (*Figure 6—figure supplement 1*). The data showed that the T/F07 H173- and ΔDSV-induced antibodies reacted with autologous T/F07 Env proteins, and with clades C-1086 and AE244 HIV Envs that encode C-strand sequence closely resembling the T/F07 sequence. On the other hand, the Y173 and Y173.ΔDSV induced antibodies reacted broadly with all the proteins tested, and more strongly overall than the T/F07-induced sera as evident from the heat map and the summarized binding response (*Figure 6B–C*). From these, it is reasonable to conclude that the H173 immunogen induced antibodies with narrow specificity to C β-strand whereas the Y173 immunogen induced more broadly reactive antibodies.

Next, we tested whether the narrowly specific H173-induced antibodies are similar to CH58 and CH59 mAbs derived from RV144 vaccinees. Since the immunogen used in RV144 trial contains the H173 allele and that the CH58 and CH59 mAbs specifically recognized H173 C β-strand epitope but not the Y173 variant, we hypothesized that the H173 V2, but not the Y173 V2, might induce CH58/59-like antibody responses. This was tested by blocking assays using CH58 and CH59 mAbs. The H173 gp140 Env antigen was coated on ELISA plates and were exposed to H173 or Y173 mice sera after first blocking by treatment with CH58 or CH59 mAbs. Remarkably, the H173 sera, but not the Y173 sera, with or without ΔDSV, showed significant reduction in binding (*Figure 6D–E*). Furthermore, since CH58 and CH59 binding is sensitive to mutation at D180 residue that is adjacent to C β-strand (*Liao et al., 2013*), we evaluated its binding. Consistent with the above antibody blocking data, binding of H173-induced antibodies but not of Y173-induced antibodies was significantly more sensitive to D180A mutation (*Figure 6F–I*). Overall, these data suggest that the presence of histidine at position

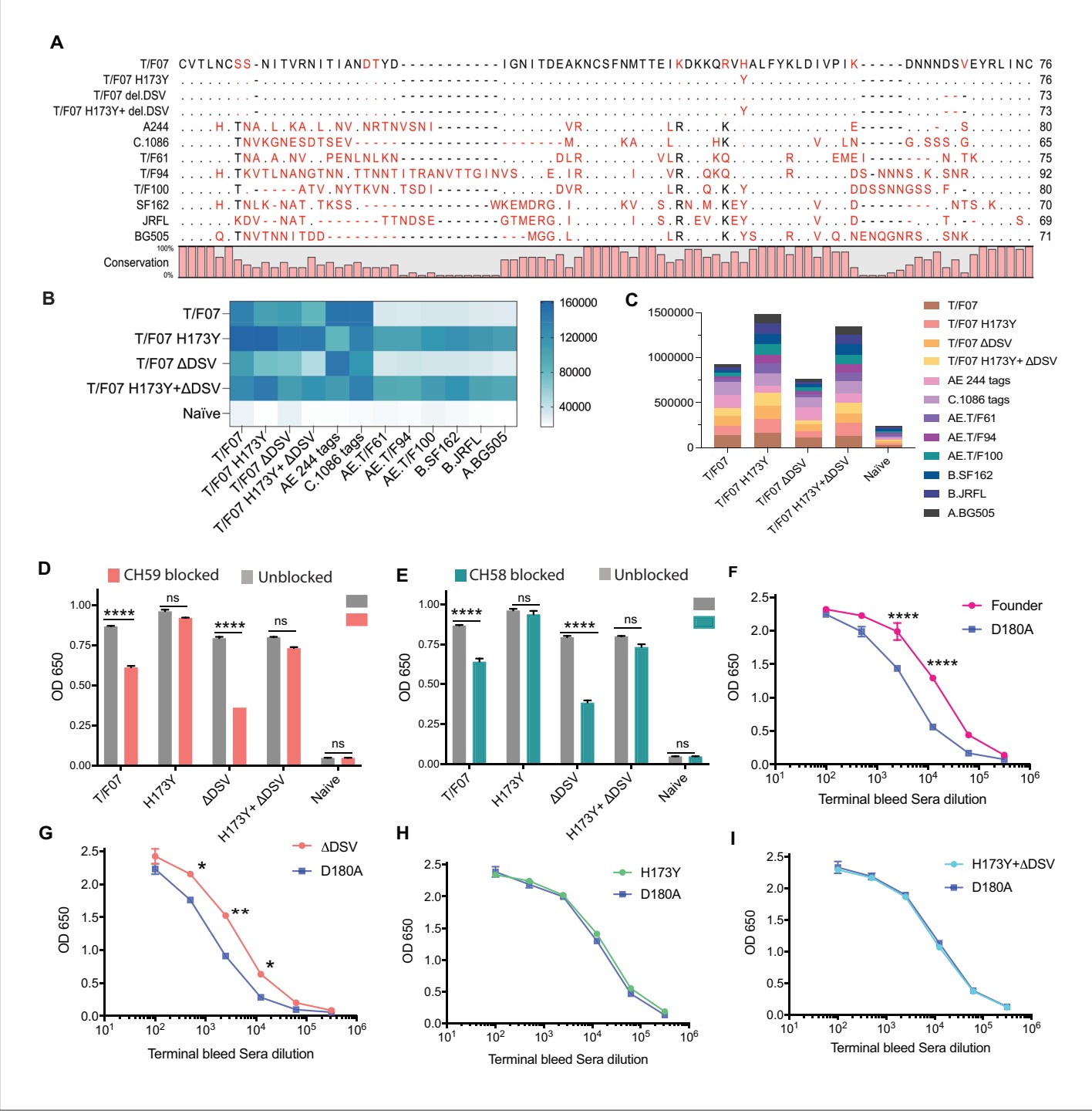

**Figure 6.** The H173 and Y173 V2-variants induce distinct antibody responses. (**A**) Sequence alignment of V1V2 region of all the antigens used in the cross-reactivity ELISA experiment. The sequences are compared with T/F07 V1V2 (top-most) sequence. T/F07 matching residues are shown as dots and different residues are highlighted in red. Difference in the lengths of hypervariable loops is shown by dashed lines (gaps). Degree of conservation is depicted graphically at the bottom of the sequence alignment. (**B**) Heat-map showing cross-reactivity for each immunized group. The map is generated based on AUC values calculated from the binding curves shown in *Figure 6—figure supplement 1* for each immunized group. The antigens used in the binding experiment are labeled horizontally at the bottom of the heat-map. The immunization groups are labeled vertically on the left-side of the heat map. The color gradient scale on the right side shows the degree of reactivity with corresponding numerical values. (**C**) Cumulative AUC values are plotted to display the summarized or total response of each immunization group towards all the antigens, calculated from the binding curves shown in *Figure 6—figure supplement 1*. Each antigen is depicted as a small rectangle colored with respect to the key provided at the side of the

*Figure 6 continued on next page*

*Figure 6 continued*

graph. Binding to each antigen corresponds to the area of rectangle in the bar graph. (**D–E**) In an antibody blocking assay, significant reduction in sera reactivity was observed in gp140 T/F07 coated wells pre-incubated/blocked with purified mAbs, CH59 (D, orange), and CH58 antibodies (E, teal), for gp16-T/F07 or ΔDSV (H173 variants) but not gp16-H173Y or H173Y+ΔDSV (Y173 variants) immunized mice groups, compared with the unblocked wells (gray). Titrated and optimized sera dilution was used for each group in this assay performed with three technical replicates. (**F–I**) Sensitivity to D180A mutation. Binding curves showing reactivity of gp16-T/F07 (**F**), H173Y (**G**), ΔDSV (**H**) and H173Y+ΔDSV (**I**) immunized sera to respective autologous T/F07-gp140 (with matching V2 mutations) coating antigens (color-coded curves) versus T/F07 gp140-D180A mutant (blue curves). Binding is determined by ELISA performed with two technical replicates. Absorbance (OD 650 nm) readings were used to generate binding curves. p-values were determined through unpaired t-test, ****=p < 0.00001, ***=p < 0.0001, **=p < 0.001 and *=p < 0.01.

The online version of this article includes the following source data and figure supplement(s) for figure 6:

**Source data 1.** Data used to generate graphs in *Figure 6B-I*.

**Figure supplement 1.** Antibody breadth analysis of the H173 and Y173 V2 variants-immunized mice sera.

**Figure supplement 1—source data 1.** Data used to generate graphs in *Figure 6—figure supplement 1*.

173 favors induction of CH58/59-like antibodies with narrow specificity as were elicited by the RV144 vaccinees.

The above datasets show that the mouse H173-induced antibodies, like their human counterparts, are narrowly specific to the autologous C β-strand and hence sensitive to sequence variation. Conversely, the Y173-induced antibodies are broadly reactive, conformation-dependent, and cross-reactive to diverse V2 domains that differ in length, sequence, and glycosylation. Furthermore, these antibodies also tolerated variations in C β-strand sequence not only at 173 position but also at other critical positions such as K168 or K169 (*Rolland et al., 2012*; *Liao et al., 2013*).

## Structural analyses indicate conformational switching in virus escape

To determine if the mutational switch involved in virus escape might be due to a conformational switch in the C β-strand, we performed Molecular Dynamics (MD) simulations and structural modeling analyses of H173- and Y173-V1V2 domains. For MD simulations, we first modeled H173- and Y173-V1V2 domains using MODELLER 9v7 (*Sali and Blundell, 1993*) based on our recently published cryo-EM structure of CRF_AE T/F100 HIV-1 Env trimer as template (PDB ID: 6NQD), due to their extensive sequence similarity (87.6%) (*Ananthaswamy et al., 2019*). Both these V1V2 domains assumed β-stranded conformation matching the template. The V1V2 domains were extracted to run simulations using GROMACS 5.1.2 (*Van Der Spoel et al., 2005*) and trajectories were produced for 100 ns with 2fs time step. We then explored how H173 and Y173 strand V1V2 models undergo changes in conformation over time.

In H173/Strand but not in Y173/Strand trajectory, helix content increased over time, though the β-pairing within the domain remained stable. In contrast to H173/Strand trajectory, Y173/Strand trajectory displayed a dynamic and fluctuating β-strand content (*Figure 7A*, left). We further investigated how the V1V2 conformation would change if the C strand was a helix in the initial conformation. To construct a helix initial model, helical restraints were put on residues, 167–176 (DKKQRVH/YALF), following the helical conformation of the published crystal structure of V2/C-strand peptide (PDB ID: 4HPO). The helical structures were also modeled using MODELLER 9v7. Different from the strand conformation simulations, trajectories with helix initial conformations showed unstable trajectories for both H173 and Y173 V1V2 domains. The overall β-strand content increased around 30 ns in both the simulations (*Figure 7A*, right). However, while helix in Y173/Helix model was fully unwound during simulation, one turn encompassing residues 166–171 remained in H173/Helix until the end of the trajectory (*Figure 7B–C*). Furthermore, the unwound region of H173/Helix but not Y173/Helix model could still engage in β-pairing. Overall, the MD simulations data suggested distinct conformational dynamics for H173 and Y173 V1V2 domains with the latter being relatively more dynamic. Furthermore, while the conserved β-sheet conformation is thermodynamically favored and hence predominates for V1V2 domain, H173/C-strand region could tolerate helical constraint owing to the stable β-pairing in the rest of the domain.

Since the template-based modeling tends to get biased towards the template structure, we next performed structural modeling of H173- and Y173-V1V2 domains using a modeling tool QUARK (*Xu and Zhang, 2012*) that generates *ab initio* structure predictions based on physical principles rather than previously resolved structures as templates. Five 3D models depicting possible conformations

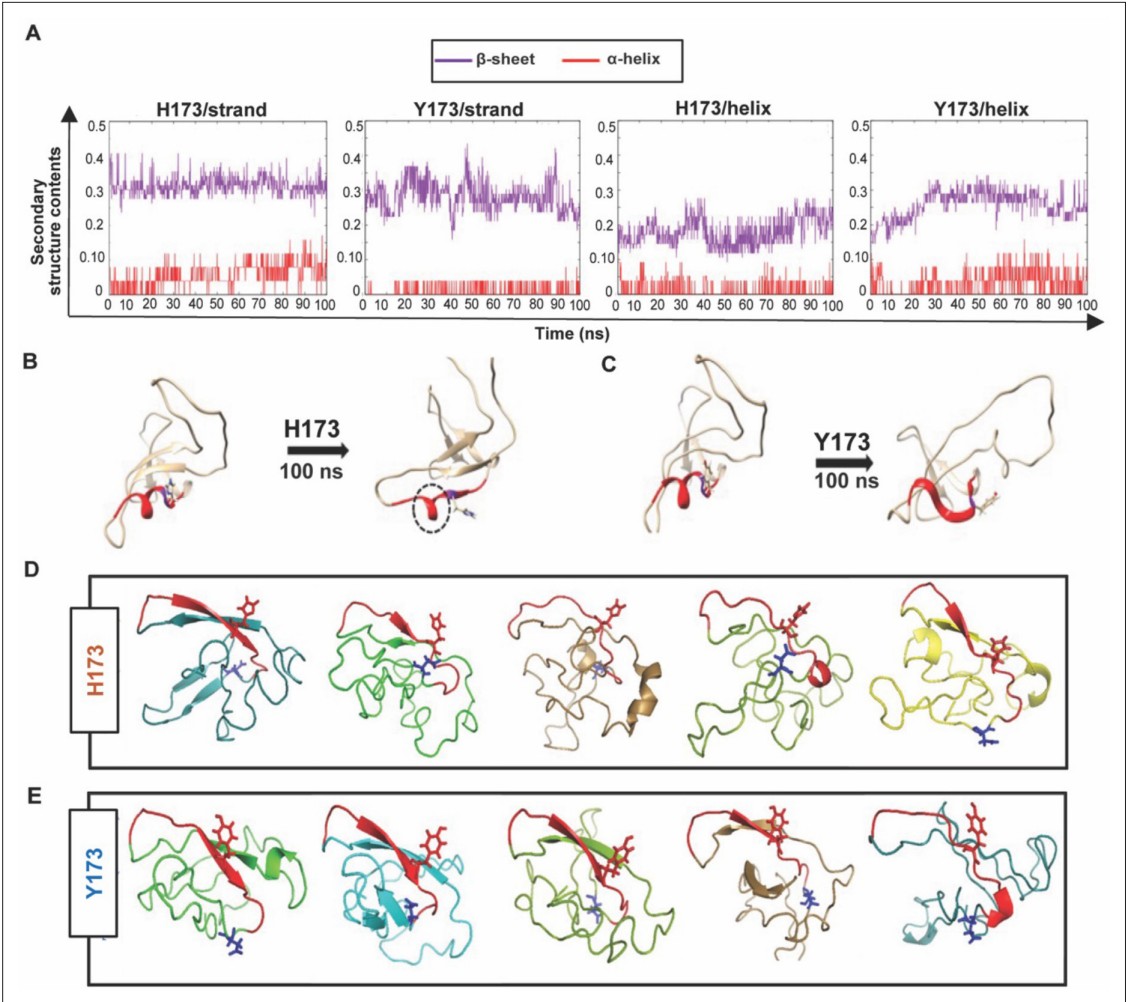

**Figure 7.** Structural analyses of H173 and Y173 variant V1V2 domains. (**A**) Secondary-structure content variation as a function of time in nanoseconds (ns) initiated with V2/strand conformation (left) and V2/helix conformation (right) of H173 and Y173 V1V2 models during MD simulation. The β-sheet content (purple) and α-helix content (red) changes are depicted over the 100 ns time-course of simulation. (**B–C**) Initial (left) and final (right) structures of the V2/helix trajectory for H173 (**B**) Y173 (**C**) V1V2 models. Histidine and tyrosine side chains are shown in the model with C strand highlighted in red. A portion of the helix remains stable (encircled by black dashed lines) at the end of 100 ns simulation time in H173/helix trajectory. (**D–E**) Ab initio structure modeling of H173 and Y173 variant V1V2 domains. Five models were generated through QUARK modeling tool for H173-V1V2 (**D**) Y173-V1V2 (**E**). C strand is colored in red with sidechain shown for residue 173. Sidechain for residue 180 (critical for helix/coil V2-conformation recognizing CH58/59 like antibodies) is shown in blue.

were generated for each V1V2 variant based on replica-exchange Monte Carlo simulation under the guide of an atomic-level knowledge-based force field. It was found that only one of the five H173-V1V2 models depicted C strand as β-strand, while the rest showed a coil plus a short β-strand or a full coil conformation (*Figure 7D*). In contrast, four of the five Y173-V1V2 models depicted C strand in β-stranded conformation while only one model showed this region to assume coil conformation (*Figure 7E*). Overall, these models indicate that H173Y mutation causes a structural change in the C β-strand thereby altering the conformational dynamics of the V1V2 domain. Furthermore, notably, models of both the variants depicted the C strand as β-strand to some degree which represent the conserved Greek key motif captured by all the resolved Env trimer structures. However, the degree of β-stranded character was identified to be much higher in Y173 variant than in H173 variant. Hence, it is plausible that the observed antibody evasion and virus escape that occurred in participant 7 was due to a structural transition in this V2 epitope, from helix to β-strand owing to H173Y mutation.

## Combinatorial V2 immunogen library construction rationale

The H173Y mutation leading to structural transition, viral escape, and induction of broadly reactive antibody species provided a conceptual basis to design an ensemble of V2-conformation variants, which could further determine if H173 and Y173 immunotypes can exhibit any effect on the V2 conformation and antibody responses with other V2 mutations. Furthermore, we hypothesized that combinatorial immunogen libraries might further broaden antibody responses which might have implications for vaccine design.

For designing these combinatorial immunogen libraries, first, we sought to identify the potential escape sites in the semi-conserved C β-strand which is previously reported to be conformationally polymorphic and a prime target of V2-directed antibodies. HIV sequence database was explored to extract 100 Env sequences from each of the major geographically prevalent HIV subtypes including A, B, C, and AE. These sequences were individually aligned using CLC Main Workbench and the partly conserved C strand sequence alignment was extracted to generate a consensus logo (*Figure 8A*). Subsequently, sites of highest variability were identified in the consensus logogram with the rationale that a less conserved or highly variable site is likely to be linked to viral escape. This resulted in the prediction of four residues-at amino acid positions 166, 169, 170, 171 as highly variable. Then, primers were designed such that the most common variants in the natural HIV-1 population, lysine (K), arginine (R), or glutamine (Q), were incorporated at each of these positions through controlled mutagenesis approach to generate a combinatorial library (*Figure 8B*). The resultant library encompassed 54 V2-variants possibly representing the V2 conformations prevalent in the HIV population. The rationale behind choosing naturally selected mutations is not only to represent natural diversity of HIV-1 but to also ensure the structural and functional integrity of the V1V2 domain, thereby presenting only the most relevant epitope diversity to the immune system.

Four different combinatorial gp16-V1V2 nanoscaffold libraries were constructed. Three of these are in the background of H173, Y173, and Y173.ΔDSV templates. The fourth library has a V1 loop deletion as this deletion has previously been shown to modulate the immunogenicity of the Env protein and the subsequent antibody responses (*Saunders et al., 2005*; *Sanders et al., 2000*; *Silva de Castro et al., 2021*). For this library, we used the Y173 template in which a 15-amino acid residue mutational hotspot in the V1 loop (SNITVERNITIANDTYD) was replaced with a flexible linker (AGGAS) optimized through in silico structural modeling such that it will have minimal impact on the V1V2 backbone conformation. This Y173.ΔV1 library is also supposed to eliminate certain immunodominant residues in the V1 loop and enhance V2-directed antibody responses (*Silva de Castro et al., 2021*). These four combinatorial libraries plus the two original H173 and Y173 V2 immunogens as controls, were all expressed as gp16 nanoscaffolds in GnTi cells, and the recombinant proteins were purified. To minimize epitope distraction, a cleavable HRV3c protease site was engineered into each of the constructs and the strep-tags were removed from the purified proteins by treatment with the protease. The protease was then separated from the scaffolds by size-exclusion column chromatography (*Figure 8—figure supplement 1*). The pure, tag-less, V1V2 nanoscaffolds were then used for mouse immunizations and induction of V2-specific antibodies was evaluated.

## Combinatorial immunogens broaden V2 antibody responses

BALB/c mice (4 mice/group) were immunized at wks 0, 3, 6, and 9 with pure V2 immunogen libraries as described above (*Figure 5A*). Naïve mice (PBS/no antigen) and mice immunized with gp16-scaffold alone (without V2) served as negative control groups. Sera were collected after the final boost and analyzed by a series of immunological assays.

V1V2-specific antibody titers were quantified by ELISA using the respective purified proteins as coating antigens. V1V2 antibodies were detected after prime immunization and enhanced by several fold with each successive boost for all the groups except for the negative control groups where no V2-specific responses were detected. The terminal bleed sera which had the maximum antibody titers were then used for detailed epitope specificity studies (*Figure 8—figure supplement 2*).

To determine the recognition breadth of antibody responses, a series of ~20 heterologous recombinant gp140 and gp120 Env proteins from diverse HIV subtypes A, B, AE, and C were used as coating antigens for ELISA assays. Many of these were purified from GnTi cells while some were obtained from NIH Reagent Program (*Figure 8—figure supplement 3A–C* and *Table 1*). These Env proteins were selected based on the differences in the composition of the V1V2 domain C β-strand with respect

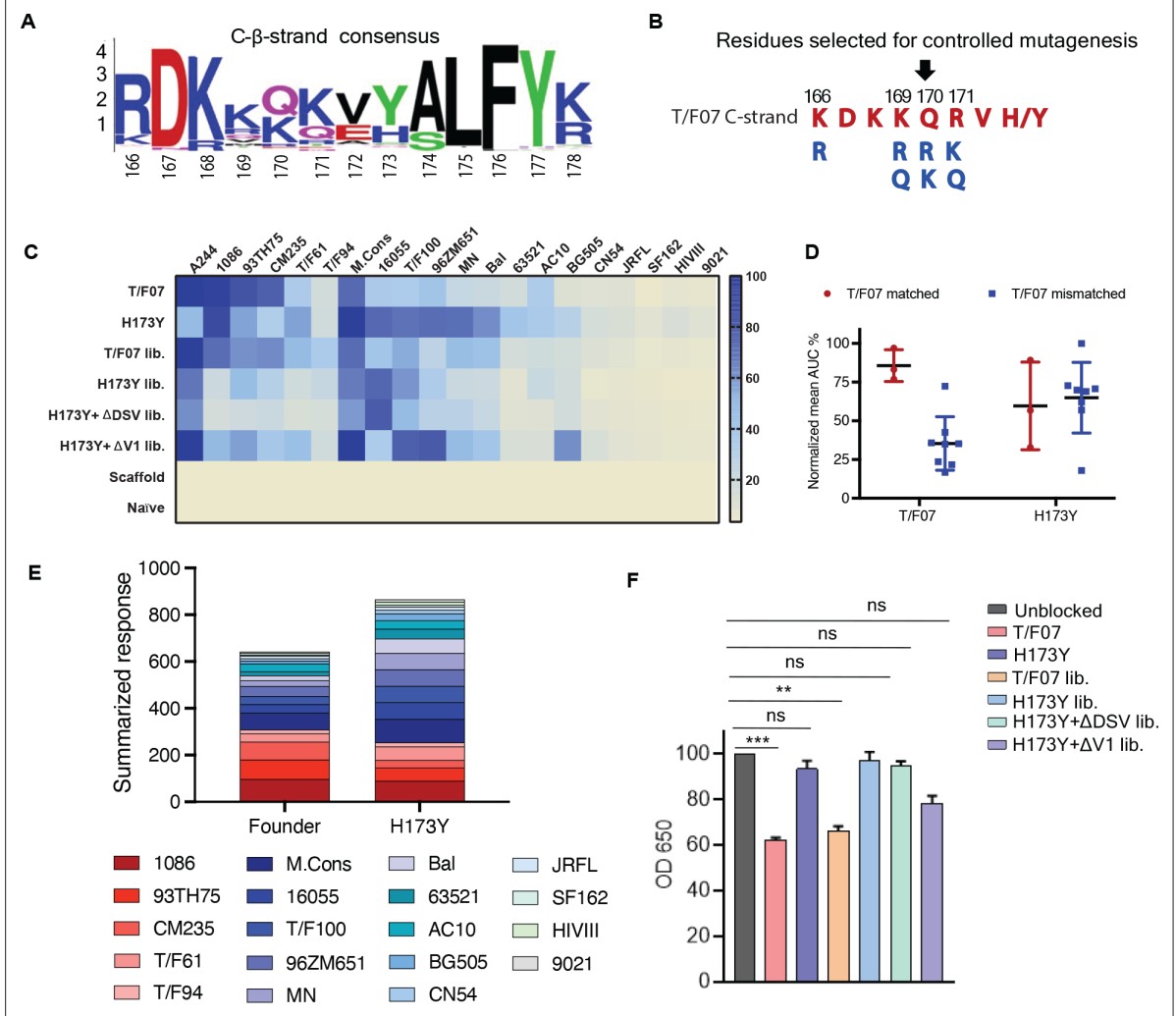

**Figure 8.** Combinatorial V2-immunogens induce broad antibody responses. (**A**) Consensus logo of C β-strand (residue 166–178) of the V1V2 region generated using 100 *env* sequences each of clades A, B, C, D, and AE. The size of the letter depicting the residue in the logo is proportional to its conservation across viral sequences under analysis. (**B**) Four highly variable residue positions (166, 169, 170, and 171) were selected for combinatorial mutagenesis with additional substitutions shown in blue to generate V2 combinatorial libraries. (**C**) Heat-map showing binding of sera from different immunized mice groups to diverse HIV-1 Env antigens. Percent AUC values derived from the binding curves and normalized to autologous antigen binding response were used to generate the heat-map. The antigens used in the binding experiment are labeled horizontally at the top of the heat-map. The immunized mice groups are labeled vertically on the left side of the heat-map. The color gradient scale on the right side shows the degree of reactivity with corresponding AUC percent values. (**D**) Effect of C strand sequence variation on the binding of T/F07 and H173Y immunized groups' sera. Scatter plot showing significant difference in the reactivity of gp16-T/F07 but not H173Y immunized sera with Env antigens having C strand sequence closely matching (red circles) versus mismatching (blue squares) to T/F07. Normalized mean AUC (%) values are plotted on the y-axis estimated from the binding curve for each antigen represented as sphere or square in the graph (each circle or square represents different antigen). (**E**) Cumulative AUC values are plotted for T/F07 and H173Y groups to display the summarized or total response from all the diverse antigens in the library. Each antigen is depicted as a small rectangle colored with respect to the key provided at the bottom of the graph. Binding to each antigen corresponds to the area of rectangle in the bar graph. (**F**) CH58 antibody blocking assay data. Significant reduction in sera reactivity was observed for T/F07 and T/F07 library groups when gp140 T/F07 coated wells in duplicates were pre-incubated/blocked with purified mAb, CH58 (grey) compared to unblocked (pink). No statistically significant inhibition of binding signal was observed for any H173Y-based immunogen groups. p Values determined using unpaired t-test, **=p < 0.01 and ***=p < 0.001 (significant difference).

The online version of this article includes the following source data and figure supplement(s) for figure 8:

**Source data 1.** Data used to generate graphs in *Figure 8C-E and F*.

**Figure supplement 1.** Purification of tag-free V1V2 immunogens.

**Figure supplement 2.** V1V2-specific binding responses in the terminal bleed sera.

**Figure supplement 2—source data 1.** Data used to generate graph in *Figure 8—figure supplement 2*.

*Figure 8 continued on next page*

*Figure 8 continued*

**Figure supplement 3.** Recombinant construction and purification of diverse HIV-1 Env proteins to determine cross-reactive responses and breadth.

**Figure supplement 4.** Breadth analysis of V2 combinatorial library immunogens using heterologous Env antigen library.

**Figure supplement 4—source data 1.** Data used to generate graphs in *Figure 8—figure supplement 4*.

to length and sequence identity, and additionally, glycosylation of their hypervariable V1 and V2 loops (*Figure 8—figure supplement 3D*). All these recombinant proteins were used as coating antigens to determine the cross-clade antibody responses generated by different combinatorial libraries (*Figure 8—figure supplement 4*).

Consistent with the first mouse study, the Y173-induced antibodies showed greater cross-reactivity than the H173-induced sera, as shown by their binding to majority of the heterologous Env proteins. In contrast, H173-induced sera strongly reacted with Env antigens from A244, 93TH75, 1086, and CM235 strains that contained similar C β-strand sequence as the T/F07 virus, while showing moderate to poor reactivity with the rest of the Env antigens as depicted in the heat map (*Figure 8C*). Upon grouping the antigens into T/F07 matched and mis-matched C β-strand sequence, H173 antibody responses were more sensitive to mismatches in the C β-strand region than the Y173 antibodies

**Table 1.** List of reagents obtained from the NIH Reagent Program.

| Catalog # | Description |
|---|---|
| 4961 | HIV-1 BaL gp120 recombinant protein |
| 7749 | HIV-1 CN54 gp120 recombinant protein |
| 10080 | HIV-1 96ZM651 gp120 recombinant protein |
| 11556 | HIV-1 JR-CSF Fc-gp120 recombinant protein |
| 11784 | HIV-1 IIIB gp120 recombinant protein |
| 12063 | HIV-1 UG037 gp140 recombinant protein |
| 12064 | HIV-1 CN54 gp140 recombinant protein |
| 12569 | AE.A244 D11 gp120 recombinant protein |
| 12570 | B.MN D11 gp120 recombinant protein |
| 12571 | B.9021 D11gp120 recombinant protein |
| 12572 | B.6240 gp140C recombinant protein |
| 12574 | B.63521 D11 gp120 mutC recombinant protein |
| 12576 | M.CON-S D11 gp120 recombinant protein |
| 12581 | C.1086 gp140C recombinant protein |
| 13055 | HIV-1 AC10.29 gp120 Avi His recombinant protein |
| 13342 | HIV-1 93TH975 gp120 recombinant protein |
| 12567 | HIV-1 Env V1V2 recombinant protein (AE.A244 V1V2 tags) |
| 12568 | HIV-1 Env V1V2 recombinant protein (C.1086 V1V2 tags) |
| 8660 | HIV-1 96ZM651.8 gp140 optimized expression vector |
| 12806 | HIV-1 CM235 gp120 expression vector |
| 12957 | HIV-1 AC10.29 gp120 Avi His optimized expression vector |
| 13348 | HIV-1 BaL gp120 His expression vector |
| 13349 | HIV-1 93TH975 gp120 His expression vector |
| 13350 | HIV-1 CN54 gp120 His expression vector |
| 12551 | CH59 mAb |
| 12550 | CH58 mAb |

(*Figure 8D*). Cumulative response towards all the antigens was also compared for H173 and Y173 groups by plotting the summarized response based on AUC values (%) (*Figure 8E*). These data clearly showed that with single residue change, H173Y, there occurred significant improvement in recognition of diverse Env antigens and hence breadth of V2 reactivity.

The combinatorial H173-induced antibodies, like the H173-induced antibodies, were significantly inhibited by CH58 antibodies while no significant inhibition was observed for sera induced by Y173 or Y173 combinatorial groups (*Figure 8F*). These data suggest that both these immunotypes play a dominant role in influencing the conformation even in the context of a library of V2 variants. Furthermore, since CH58 recognizes V2 region as α-helix, it is consistent with our analyses described above that the H173 immunotype has more propensity for helical conformation over β-strand (*Liao et al., 2013*).

Finally, unlike the H173 group, the combinatorial H173 group showed relatively broader responses, with much improved binding to Env antigens such as T/F61, T/F94, T/F100, MN, and Bal. Hence, H173 library-based vaccine candidates led to enhanced recognition of other Env proteins. With respect to Y173 and Y173.ΔDSV combinatorial libraries, strong binding was seen with all Y173 antigens and moderate reactivity towards H173 bearing antigens. Notably, the sera induced by Y173.ΔV1 combinatorial library showed strong overall binding to both H173 and Y173 antigens. Though the breadth of reactivity for this group was comparable to a non-combinatorial Y173 group, it represented a distinct binding profile because of its stronger reactivity to most of the Env antigens such as A244, T/F100, 96ZM651, and BG505 than the H173Y group containing the V1 loop (as shown in *Figure 8C*). Thus, the Y173.ΔV1 combinatorial immunogens containing both the Y173 mutation and V1 loop deletion could be an effective design for broader and stronger cross-reactive V2 antibodies. Overall, this combinatorial approach demonstrates that accommodating putative escape mutations with either histidine or tyrosine at position 173 have differential effect on V2 conformation and antibody responses.

## Discussion

The modest 31.2% efficacy of the only successful RV144 HIV vaccine trial in Thailand was correlated with the induction of V2-directed antibodies (*Haynes et al., 2012*; *Rao et al., 2013*; *Liao et al., 2013*; *Kim et al., 2015*). Analysis of breakthrough infections of vaccinees showed mutations in the V2 domain of the circulating viruses, presumably selected for their ability to survive under the host immune pressure (*Rolland et al., 2012*; *Moodie et al., 2022*). If the vaccine-induced antibody responses had increased breadth, vaccine efficacy would have been greater and it would have also minimized the emergence of viral escape mutants. Hence, increasing the breadth of vaccine-induced antibody responses is a critical goal of HIV-1 vaccine design. Here, we report detailed analyses of a remarkable genetic shift of a transmitted/founder HIV-1 virus through a single amino acid substitution in the V2 domain to escape strong host immune pressure that also resulted in increased breadth and cross-reactivity toward diverse HIV-1 V2 domains and envelope proteins.

Phylogenetic analyses of T/F viruses in longitudinal samples of RV217 ECHO trial identified a complete genetic shift of virus population in participant 7 at 24 wk post-infection. The shifted viruses carried a histidine to tyrosine substitution at position 173 of the semi-conserved C β-strand of the V2 domain. This shift coincided with a wave of H173-epitope specific antibodies produced by the host immune system. That the Y173-epitope showed no detectable reactivity to these antibodies while the H173 epitope reacted strongly demonstrates that the Y173 substitution represents an escape mutant selected in response to the strong immune pressure. Furthermore, the Y173 epitope either did not bind, or bound poorly, to CH58 and CH59 mAbs isolated from RV144 vaccinees, which are also directed to the same C β-strand epitope and recognize H173 as a critical residue (*Liao et al., 2013*). In fact, two of the three vaccine immunogens (A244 and 93TH023) used in the RV144 trial had histidine at 173 position, suggesting the H173 specificity of these vaccine induced CH58 and CH59 antibodies (*Robb et al., 2012*).

Neither the participant 7 T/F sera, nor the RV144 vaccine sera or the CH58 and CH59 mAbs, exhibited strong virus neutralizing activity, but the latter exhibited strong ADCC activity which correlated with reduced infection in immune correlate analysis (*Mayr et al., 2017*; *Bonsignori et al., 2012*; *Bradley et al., 2017*; *Pollara et al., 2014*). Similarly, in our current study, the Y173 escape mutant which also consisted of an additional DSV tripeptide deletion in the adjacent variable V2 loop showed significant resistance to ADCC mediated killing when compared to the H173 epitope. The DSV

deletion alone otherwise had no effect on the binding of the C β-strand epitope to antibodies as evaluated by ELISA or SPR binding assays, yet it was co-selected along with the Y173 mutation. This would argue that the strong immune pressure exerted by the host might be due to the ADCC activity of the elicited antibody responses, consistent with the previous reports suggesting the importance of ADCC responses in HIV-1 infected individuals particularly the elite controllers, and the protection imparted by vaccines in non-human primates (*Bonsignori et al., 2012*; *Excler et al., 2014*; *Bruel et al., 2016*; *Dupuy et al., 2019*; *Mabuka et al., 2012*; *Isitman et al., 2016*; *Pollara et al., 2013*).

Intriguingly, a second wave of antibodies with greater breadth emerged following the H173 to Y173 genetic shift and the essential disappearance of the H173-specific first wave antibodies. Unlike the latter, the second wave antibodies bound equally well to both H173 and Y173 C β- strand epitopes, as analyzed by multiple assays including the sensitive SPR assay. Notably, an examination of Env trimer structures show that the H173 residue is at the center of the C β-strand that contains hydrophilic and solvent-exposed residues flanking on one side and hydrophobic and buried residues on the other side (*Pan et al., 2015*). A H173Y mutation could therefore have significant structural consequences. Previous X-ray structures showed that the H173 peptide epitope assumes an α-helix when bound to CH58 and CH59 antibodies as opposed to a β-strand in the envelope proteins with tyrosine at residue 173 (*Pan et al., 2015*; *Liao et al., 2013*; *Ananthaswamy et al., 2019*). Furthermore, prevalence of either H or Y residues at this position among numerous Env sequences in the database argues for functional importance of this site in HIV-1 evolution. Thus, it appears most likely that the H173Y genetic shift in T/F07 HIV-1 virus may have caused a significant structural/conformational change in the C β-strand epitope, which led to total escape of H173-specific immune pressure while inducing the second wave antibodies with distinct specificity and greater breadth.

Remarkably, the differential immunological responses observed in the human immune system were recapitulated in a mouse model. First, the gp16-V1V2 nanoscaffolds elicited antibody responses in mice that recognized the respective V2 epitopes in the context of gp120 or gp140 envelope protein structures. This is significant not only because antibodies induced by other scaffolded V2 domains do not bind well to the envelope proteins but also that it indicates that the gp16-scaffolded V2 epitopes are displayed in a native-like conformation. Second and more important, the H173 and Y173 variants induced antibodies with distinct specificities, similar to that observed in the human system. While the H173-induced antibodies strongly reacted with the autologous peptides but not with the Y173 peptide, the Y173-induced antibodies exhibited broad reactivity to both the peptides. This differential response was also observed in their reactivity towards diverse HIV-1 Env proteins. While the H173 antibodies were sensitive to mismatches or sequence diversity in the C β-strand, the Y173 antibodies showed cross-reactivity to a variety of Env proteins. Third, the H173 antibodies but not the Y173 antibodies competed with the RV144 CH58 and CH59 mAbs for binding to the C β-strand epitope.

The picture that emerges from the above data, combined with the previously reported studies, is that the C β-strand region is conformationally dynamic and probably undergoes a structural transition when H173 changes to Y, from a β-strand conformation as part of conserved Greek motif to an α-helix/helix-coil conformation (*van Eeden et al., 2018*; *Liao et al., 2013*; *Wibmer et al., 2018*). Our MD simulations and *ab initio* structural modeling of H173 and Y173 V1V2 domains further suggest that the H173 region is rigid and assumes a helical/coil conformation whereas the Y173 domain is more dynamic, preferring to be β-stranded and part of a β-sheet core. Thus, it is imperative to include both these structural forms in any vaccine design for increasing the breadth of antibody responses. In fact, a pentavalent vaccine containing five Env proteins including both the H173 and Y173 variants gave better protection in non-human primates when compared to single immunogen (*Bradley et al., 2017*).

Given the above considerations, we rationalized that for a vaccine to be highly effective, it should not only contain the H173 and Y173 variants, but additionally it should also include commonly found mutations that could fine-tune the C β-strand conformation, some of which might have arisen as escape mutants against host immune pressures in past infections. Furthermore, presenting these variants as gp16 scaffolds would be ideal because this would eliminate nonspecific distraction to other nonessential epitope sites of the Env but also that the gp16-scaffolded V1V2 domains, as discussed above, elicit antibodies that recognize the C β-strand epitopes in a native context. Therefore, a combinatorial immunogen design was developed by including both H173 and Y173 variants, each in addition carrying combinations of the commonly found substitutions in the C β-strand as informed

by sequence analyses. These combinations, a total of 54 variant epitopes in each library, in some respects might mimic the natural V2 conformations of HIV-1 viruses humans are exposed to at the site of entry. Assessment of the antibody responses in the mouse model showed that, indeed, some of these libraries significantly enhanced the immune breadth when compared to the respective single immunogen controls. However, consistent with the human responses, the breadth is greater in the context of Y173 when compared to H173, particularly when Y173 was combined with a deletion of V1 loop. The greater breadth and cross-reactive responses resulted from the Y173-ΔV1 library was most likely due to reducing the immunodominance of V1 epitopes as well as generation of breadth favoring conformational variants generated by Y173 switch. Consistent with these data is the recent report that showed that the responses directed to V1 loop interferes with binding of protective V2-directed antibodies to Env and promotes virus acquisition in SIV vaccinated macaques (*Silva de Castro et al., 2021*). On the other hand, the H173 combinatorial libraries induced antibodies similar to CH58 mAb as shown by the CH58 blocking data. Induction of such responses despite additional mutations in the C β-strand argues for a dominant role the H173 residue plays in assuming a helical conformation that is recognized by the CH58 antibody (*Liao et al., 2013*).

In conclusion, our studies show that residue 173 and the presence of histidine or tyrosine at this position strongly influence the local conformation of V2 domain and this conformational dynamism might lead to virus escape and breakthrough infections. This could potentially be curtailed by combinatorial vaccine designs such as the Y173-ΔV1 library of immunogens that can display a set of dynamic and flexible conformations and have the potential to generate broad antibody responses. This type of escape and conformational variations could also be obtained on a larger scale by AI-based structural modeling and machine learning using the vast HIV sequence database from various clinical trials. These approaches in future could lead to highly effective HIV-1 vaccines with enhanced efficacy and greater ability to block diverse virus transmissions.

# Materials and methods

## Viral load analysis

Viral loads were determined in participant 7 longitudinal samples from wk 0 up to wk 144 as described previously (*Trinh et al., 2019*; *Robb et al., 2016*). Briefly, viral RNA was isolated from cell-free plasma using the QIAamp viral RNA isolation kit (QIAGEN). Quantitative reverse transcription-PCR was conducted in a two-step process. First, RNA was reverse transcribed followed by treatment with RNase H (Stratagene) for 20 min at 37 °C and then cDNA was quantified using specific amplification primers, dyes, and probes. All reactions were carried out on a 7300 ABI real-time PCR system with TaqGold polymerase (Applied Biosystems) according to the manufacturer's protocols.

## Single genome amplification (SGA)-derived envelope sequencing

SGA sequencing was performed at viral sequencing core in Walter Reed Army Institute of Research (WRAIR) as described previously (*Trinh et al., 2019*; *Keele et al., 2008*). Briefly, viral RNA was extracted from the plasma of the infected RV217 participants using the QIAamp Viral RNA Mini Kit (QIAGEN, Valencia, CA, USA) and complementary DNA (cDNA) was synthesized using the SuperScript III RT kit (Invitrogen/Thermo Fisher Scientific, Waltham, MA, USA) following the manufacturer's instructions. cDNA was amplified as a full genome or 2 half genomes overlapping by 1.5 kb as previously described using SGA strategy, which was then end-point diluted in 96-well plate, such that to yield less than 30% amplification product. Env specific primers were used to amplify *env* gene from the HIV genome.

## Phylogenetic analysis

Multiple sequence alignments and construction of phylogenetic trees were done using CLC Main Workbench (ver. 7.6.1) software. SGA derived *env* sequences obtained from various time points (1-, 4-, 24 wk post-infection) were aligned with the respective T/F virus sequence using following parameters; Gap Open Cost = 10.0; Gap Extension Cost = 1.0; and Alignment Sensitivity = Very Accurate. Phylogenetic trees were constructed using Neighbor-Joining method, Jukes Cantor protein distance measure, and 100 bootstrap replicates. For mutational frequency analysis, the total number of mutations at each residue of the Env sequence was determined. Numbering of each residue is consistent with the HXB2 strain sequence used as reference. For consensus logo construction, Env sequences of

diverse subtypes were fetched from HIV sequence database, aligned using CLC Main Workbench. The logo was generated using an online tool, WebLogo (https://weblogo.berkeley.edu/logo.cgi).

## Structural modeling

3D model of a gp140-T/F07 was generated using homology modeling server, SWISS-MODEL. BG505 gp140 (PDB ID: 4TVP) trimer was used as a template. Mutations were mapped on the modeled trimers using PyMol (ver. 1.74) molecular visualization software (*Schrödinger, 2010*). *Ab initio* structural modeling of T/F07- and Y173-V1V2 domains was conducted using QUARK online tool.

## Molecular dynamics (MD) simulation

First, T/F07-H173 and -Y173 trimer models were generated by MODELLER 9v7 using T/F100 Env cryo-EM structure (PDB ID: 6NQD) as template for modeling. After predicting trimer structures, V1V2 domains were extracted and used for MD simulation. GROMACS 5.1.2 was used to run four MD simulations: H173/Strand, Y173/Strand, H173/Helix, and Y173/Helix as initial models. AMBER99SB-ILDN force field was employed for protein. Dodecahedron periodic box was solvated by TIP3P water molecules. The size of the solvation box was set to 1.0 nm from the V1/V2 domain. Sodium ions were added to neutralize the system. After solvation, steepest decent minimization with 50000 steps was applied. Then, the systems were equilibrated at 300 K during 100 ps (NVT equilibrium) and at 1.0 bar during 100 ps (NPT equilibrium). After equilibration, 100 ns MD trajectory was produced for each system. The equilibration and MD production were done with 2 fs timestep, applying LINCS algorithm.

## Recombinant plasmid constructions gp16-V1V2 scaffolds and combinatorial libraries

The gp140-TF07 *env* sequence (spanning gp120 and the gp41 ectodomain up to amino acid 664) was codon optimized and synthesized using GeneArt Strings gene synthesis (Life Technologies). This synthetic fragment was used as template to amplify T/F07-V1V2 sequence corresponding to the residues 117–206 of the Env. The V1V2 sequence was cloned into pCDNA3.1(–) mammalian expression vector engineered to harbor codon optimized bacteriophage T4 terminase, gp16 with an N-terminal Gaussia luciferase (GLuc) signal peptide for secretion of recombinant protein into the media, and a Twin Strep-tag II sequence (WSHPQFEK) for affinity purification at the C-terminus. The amplified V1V2 fragment was cloned downstream to gp16 followed by Twin-strep tags. After the construction gp16-T/F07 V1V2 clone, point mutation and/or deletion was introduced using site-directed mutagenesis to construct gp16-H173Y, -ΔDSV and -H173Y+ΔDSV V1V2 scaffolds. For the construction gp16-V1V2 combinatorial libraries, these scaffolds were used as templates. Controlled mutagenesis was performed using randomized primer pool (IDT) with desired mutations or substitutions at the selected positions in the C strand to generate V1V2 mutant libraries. Amplified V1V2 library fragments were then cloned into the same pcDNA 3.1 vector used previously but with an exception that this vector was further engineered to harbor HRV3c protease cleavage site positioned before the Twin-strep tags for tag removal. For the construction of gp16-H173Y_ΔV1 combinatorial library, H173Y template with a stretch of V1 loop residues (SNITVERNITIANDTYD) deleted and replaced by an optimized short-linker (AGGAS) was used.

## gp140 and gp120 Env clones

Codon-optimized gp140 *env* sequences (spanning gp120 and the gp41 ectodomain up to amino acid 664) from T/F07, T/F61, T/F94, T/F100, BG505, JRFL and SF162 HIV-1 viruses harboring trimer-stabilizing SOSIP mutations (*Sanders et al., 2013*) and six arginine (R6) furin cleavage site (replacing native REKR cleavage site at the junction of gp120 and gp41) were synthesized (*Binley et al., 2002*). The gp140 genes were then cloned into pCDNA 3.1(–) vector that was engineered to harbor an N-terminal Cluster of Differentiation 5 antigen (CD5) signal peptide for secretion of the recombinant proteins into the media and 8X-Histidine tag for affinity purification. Additional mutations/deletions were introduced in gp140-T/F07 using site directed mutagenesis kit (NEB). The sequences of the recombinant clones were verified through sequencing (Retrogen, Inc). The gp120 expression vectors were obtained from NIH AIDS Reagent Program (*Table 1*). The furin-expressing plasmid, Furin:FLAG/pGEM7Zf(+), was obtained from Dr. Gary Thomas (Vollum Institute, Portland, Oregon). The furin fragment from this plasmid was subcloned into pcDNA3.1(–) (Life Technologies).

## Cell lines and media

HEK293S GnTI- (ATCC CRL-3022) suspension cell line was obtained from ATCC. Cell line identity was verified by STR profiling and cells were tested to be free from mycoplasma contamination. HEK293S GnTI- cells were used for expression of HIV Env proteins and maintained in FreeStyle 293 expression medium (Life Technologies), supplemented with 1% heat-inactivated fetal bovine serum (FBS, Quality Biologicals). All cells were grown in suspension in a Multitron Pro orbital shaker (Infors HT) incubator at 37 °C in 8% $CO_2$, 80% humidified atmosphere.

## Transfection

Plasmid DNAs for transfection were purified using Plasmid Midi kit (Qiagen) as per manufacturer's instructions. Transfections were carried out as described previously (*AlSalmi et al., 2015*). Briefly, GnTi cells were grown to $1 \times 10^6$ /ml cell density for transfection. Prior to transfection, cells were centrifuged at 100 rpm for 5 min followed by full replacement of media with the fresh Freestyle293 media lacking FBS. The final cell density was adjusted to $2 \times 10^6$ / ml in half or 50% of the final volume of transfection. The cells were then placed in the shaker incubator for 1 hr at 37 °C in 8% $CO_2$, 80% humidified atmosphere. After incubation, DNA (1 µg/ml final transfection volume) was added followed by addition of linear polyethylenimine (PEI25k, Polyscience, Inc) (1 mg/ml) at a 3:1 ratio (PEI:DNA) to the cell suspension. For gp140 expression, cells were co-transfected with furin plasmid DNA to produce cleaved gp120 and gp41 subunits that then associate non-covalently to yield native Env proteins. After 12 hr of transfection, HyClone SFM4HEK293 medium (GE Healthcare) supplemented with 1% FBS (v/v) and protein expression enhancing sodium butyrate (*Reeves et al., 2002*) solution (SIGMA-ALDRICH) to a final concentration of 2 nM were added to the cells to make up to the final volume of transfection. After 5 days of transfection, the supernatant was harvested by centrifuging the cells, and filtered using a 0.2 µm filter (Corning, Inc).

## Protein purification

Secreted twin StrepTagged gp16-V1V2 proteins in the harvested and filtered supernatant were supplemented with BioLock biotin blocking solution (IBA Lifesciences) at 5 µl/ml to mask the biotin present in the supernatant. After 30 min of incubation, the supernatant was loaded onto a 1 ml Strep-Tactin column (QIAGEN) at a flow rate of 0.7 ml/min in the ÄKTA prime-plus liquid chromatography system (GE Healthcare). Non-specifically bound proteins were washed off by passing at least 20 column volumes of the wash buffer (300 mM NaCl, 50 mM Tris-HCl, pH 8) or until the absorbance reached the baseline level. Bound gp16-V1V2 proteins were eluted with StrepTactin elution buffer (5 mM d-Desthiobiotin, 300 mM NaCl, 50 mM Tris-HCl, pH 8) at a flow rate of 1 ml/min. Eluted peak fractions were buffer exchanged into 100 mM NaCl, 50 mM Tris-HCl, pH 8 buffer. Protein fractions were stored with 10% glycerol at –80 °C until use for antigenicity and immunogenicity studies. GnTi expressed His-tagged gp140s and gp120s were purified from the harvested and clarified supernatant using Ni-NTA agarose beads (Qiagen) following manufacturer's instructions.

## StrepTag removal from gp16-V1V2 immunogens

For the second mice immunization study, the Twin-StrepTags were cleaved off the immunogens using HRV3c protease. The recombinant proteins eluted after StrepTactin affinity chromatography, were buffer exchanged with 1 X HRV3C protease buffer to remove the desthiobiotin present in the elution buffer. One µL of protease was added per 20 µg of the purified protein (1:20) and incubated at 4 °C for 16 hr. Digested protein was passed twice through StrepTactin spin column (IBA). Uncleaved Strep-Tagged protein bound to the column while the flow-through containing desirable cleaved fraction was collected. The cleaved protein was then loaded onto the size-exclusion chromatography column for fractionation using 100 mM NaCl, 50 mM Tris-HCl, pH 8 buffer. Owing to a large difference in the native dodecameric gp16-V1V2 (~336 kD), HRV3C protease (47.8 kD) was separated from the final immunogens. The gp16-V1V2 fractions were pooled, concentrated and stored at –80 °C until use.

## Biochemical analyses and quantifications

Purified proteins were run on SDS-PAGE to quantify and assess for non-specific protein contamination. SDS-PAGE analyses were performed using 4–20% gradient Tris-glycine gels (Life Technologies) or home-made 12% gels in the presence of DTT (reducing conditions) or absence of DTT (non-reducing).

All gels were stained with Coomassie blue R-250 solution. Band intensities were measured using Bio-Rad Gel Doc XR +System and Image Lab software. BSA standards were used to generate a standard curve for quantification. Deglycosylation was also performed to sharpen the bands for accurate quantitation as HIV Env proteins are glycosylated and hence appear fuzzy on gels. For deglycosylation, 1 µl (500 Units) of PNGase F (New England BioLabs, Inc) was used to deglycosylate 10 µg of the protein in the presence of 5 mM DTT and mild detergents by incubating at room temperature for 1 hr according to manufacturer's recommendations.

## Western blotting

Proteins separated by SDS-PAGE were transferred to a PVDF membrane using the Trans-Blot Turbo RTA Mini PVDF Transfer Kit (Bio-Rad Laboratories, Inc). Membranes after activating with methanol were blocked with bovine serum albumin (Amresco, LLC). For Strep-Tag II detection, HRP-conjugated StrepMAB-Classical MAb (IBA Life Sciences) was used at 1:3000 dilution in PBS. Purified mAbs, CH58 and CH59 were used as primary antibodies at 1:5000 dilution in PBS and rabbit anti-human Ab HRP conjugate (Santa Cruz Biotechnology) was used as secondary antibodies at 1:10,000 dilution in PBS. Signal from HRP-conjugated antibodies was detected using Clarity Western ECL Blotting substrate (Bio-Rad Laboratories, Inc). Band intensities were measured using Bio-Rad Gel Doc XR +System and Image Lab software.

## Enzyme linked immunosorbent assay (ELISA)

### Streptactin ELISA

StrepTactin ELISA was performed to determine CH58 and CH59 binding to gp16-V1V2 proteins. These specialized plates are pre-coated with StrepTactin to capture strep-tagged antigens. Since the antigen does not directly bind to the plate surface it is maintained in native conformation that improves antibody recognition. To perform this assay, StrepTactin coated microplates (IBA Life Sciences) were coated with 1 µg/ml Strep-tagged proteins in a volume of 100 µl per well of buffer (25 mM Tris-HCl, pH 7.6, 2 mM EDTA, and 140 mM NaCl) and incubated for 2 h at 4 °C. Following three washes with PBST (0.05% Tween-20 in 1 X PBS), 100 µl of serially diluted Abs (10–0.001 µg/ml) in PBS were added to the wells and the plates were incubated for 1 hr at 37 °C. After three washes with PBST, the plates were incubated with 100 µl of rabbit anti-human Ab HRP conjugate at 1:3000 dilution in PBS for 30 min at 37 °C. The plates were then washed three times with PBST and the peroxidase substrate was added to develop the color reaction (TMB Microwell Peroxidase Substrate system, KPL). The reaction was terminated by adding 100 µl of BlueSTOP solution (KPL) and OD650 was recorded using VersaMax ELISA Microplate Reader (Molecular Devices).

## Conventional ELISA

The 96-well Nunc ELISA plates were coated with 100 ng/well antigen diluted in 1 X PBS to a concentration of 1 µg/ml, for overnight at 4 °C. After 12 hr, the plates were washed thrice with 1 X PBST (1 X PBS +0.05% Tween), followed by blocking with 5% BSA in 1 X PBS for 1 hr at room temperature (RT). After incubation, plates were washed thrice with 1 X PBST, followed by addition of 100 µL of primary antibody or serum dilution for 1 hr at 37 °C. After incubation, the plates are washed three times same as before and rabbit HRP-conjugated anti-mouse IgG (H+L) (Novex, Life Technologies) as secondary antibody was added, followed by 30 min incubation at RT. The remaining procedure is same as described above for StrepTactin ELISA.

## Antibody blocking assay

Antibody blocking assay was conducted by modifying the ELISA protocol. Briefly, 96-well Nunc ELISA plates were coated with 100 ng/well gp140-T/F07 antigen diluted in 1 X PBS to a concentration of 1 µg/ml, for overnight at 4 °C. After 12 hr, the plates were washed thrice with 1 X PBST, followed by blocking with 3% BSA in 1 X PBS for 1 hr at room temperature (RT). Plates were washed thrice with 1 X PBST, followed by addition of 100 µL of CH58 or CH59 antibody dilution (1 µg/ml) for 1 hr at 37 °C. Wells not preincubated with CH58/CH59 antibodies served as unblocked positive control. The antibodies blocked and unblocked (control) plates were washed three times same as before and incubated with optimized mice sera dilutions (1:2000) from various immunization groups for another 1 hr at 37 °C. After washing three times, the plates were incubated with anti-mouse HRP conjugated

secondary antibody for 30 min at RT. The plates were then washed followed by TMB peroxidase substrate addition (KPL) and read as described above for ELISA.

## Surface plasmon resonance (SPR) binding assay

Longitudinal plasma samples of RV217_ participant 7 were analyzed by SPR for the presence of V2-specific Abs. SPR measurements were made with a Biacore 4000 system (GE Healthcare, Uppsala, Sweden). The assay was conducted as described previously (*Trinh et al., 2019*). The subsequent procedure was conducted in a Biacore 4000 system. The immobilizations were completed in 10 mM HEPES and 150 mM NaCl pH 7.4 using a standard amine coupling kit. The CM5-S series chip surface was activated with a 1:1 mixture of 0.4 M 1-ethyl-3-(3-dimethylaminopropyl) carbodiimide hydrochloride (EDC) and 0.1 M N-hydroxysuccinimide (NHS) for 600 s. Then 20 µg/mL of each protein diluted in 10 mM sodium acetate pH 4.5 was coupled for six minutes. The immobilized surface was then deactivated by 1.0 M ethanolamine-HCl pH 8.5 for 600 s (*Trinh et al., 2012*). Spot 3 in each flow cell was left unmodified to serve as reference and to subtract the buffer shifts. After the surface deactivation, the immobilized Response Unit densities were recorded for each protein: gp16-T/F07, gp16-H173Y, gp16-ΔDSV, and gp16-H173Y.ΔDSV. Following the surface preparation, the heat inactivated sera from participant 7 were diluted (1:100) in the running buffer (10 mM HEPES, 300 mM NaCl and 0.005% Tween 20). The diluted sera were injected onto the immobilized surface for 250 s followed by a 60-s dissociation period. The antibody bound surfaced was then enhanced with 30 µg/ml secondary Sheep Anti-Human IgG (gamma chain) Antibody (Cat#: AU004.X, Binding Site, UK) for 200 s. To regenerate the bound surface, 150 mM HCl was injected for 50 s, followed by a 60-s buffer injection. All serum samples were conducted with four replicates and collected at rate of 10 Hz, with an analysis temperature at 25C. In addition, monoclonal antibody CH58 (*Tomaras et al., 2013*) (Cat#: CH58_4 A (3453), from Dr, Liao, Duke Human Vaccine Institute), which binds to linear epitope of V2 (AE) was used to monitor the stability of the immobilized gp16 V1V2 surface on the CM5 chip over time. All injections were conducted at flow rate of 10 µl/min. Data analysis was performed using Biacore 4000 Evaluation software 4.1 with double subtractions to the spot 3 (unmodified surface) and the empty buffer. The immobilized scaffold gp16 was compared with unmodified surface, there was no significant of unspecific binding; therefore, unmodified surface (spot 3) was used as reference subtraction.

For evaluating the binding of antibodies present in immunized mice sera with V2-peptides, The subsequent procedure was conducted in a Biacore 4000 system. The immobilizations were completed in 10 mM HEPES and 150 mM NaCl pH 7.4 using a standard amine coupling kit. The CM7-S series chip surface was activated with a 1:1 mixture of 0.4 M 1-ethyl-3-(3-dimethylaminopropyl) carbodiimide hydrochloride (EDC) and 0.1 M N-hydroxysuccinimide (NHS) for 600 s (*Trinh et al., 2012*). Then 1 µM Streptavidin in 10 mM sodium acetate pH 4.5 (16,600–18,500 RU) was immobilized for 720 s. The immobilized surface was then deactivated by 1.0 M ethanolamine-HCl pH 8.5 for 600 s. Spot 3 in each flow cell was left unmodified to serve as a reference. After the surface deactivation, 25 pM Nlinked biotinylated overlapping V2-peptides synthesized by JPT Peptide Technologies GmbH (Berlin, Germany) were captured onto the streptavidin immobilized chips. These overlapping linear peptides were CSFNMTTEIKDKKQRV, IKDKKQRVHALFYKLD, IKDKKQRVYALFYKLD, HALFYKLDIVPIKDNN, IVPIKD-NNNDSVEYRLINC and IVPIKDNNNEYRLINC. Following the surface preparation, the heat-inactivated sera were diluted (1:100) in 10 mM Hepes, 150 mM NaCl pH7.4. The diluted sere were injected onto the captured surface for 240 s followed by a 90 s dissociation period. The bound surfaced was then enhanced with a 360 s injection of secondary goat anti-mouse IgG antibody (AffiniPure Goat Anti-Mouse IgG (H+L), Cat#: 115-005-003, Jackson ImmunoResearch Laboratories, Inc). To regenerate the bound surface, 125–175 mM HCl was injected for 60 s. Four replicates for each peptide were collected, with an analysis temperature at 25 °C. All sample injections were conducted at flow rate of 10 µl/min. Data analysis was performed using Biacore 4000 Evaluation software 4.1 with double subtractions for unmodified surface and buffer for blank.

## GranToxiLux (GTL) antibody-dependent cell cytotoxicity (ADCC) assay

ADCC assays were performed as described previously (*Pollara et al., 2011*). Human CD4 +T lymphoblasts, CEM.NKR.CCR5 (NIAID Reagent Repository, Cat# 0099), were used as target cells to coat recombinant gp120 proteins. The amount of coating gp120 was optimized through competition by binding of the Leu3A (anti-CD4) antibody (clone SK3; Catalog no. 340133; Final dilution 1:5; BD

Bioscience, San Jose, CA, USA). Cryopreserved peripheral blood mononuclear cells, PBMCs from a healthy donor, thawed and rested overnight in R10 media, were used as effector cells (source of effector NK cells). The following day, target cells were coated with titrated amount of T/F07 and its V1V2-mutant gp120s for 75 min at 37 °C. After incubation, coated target cells were mixed with effector cells in 30:1 ratio in 96-well V-bottom plate, followed by addition on granzymeB (GzB). Finally, fourfold serially dilutions of heat-inactivated plasma samples of RV217 participant- 7, from three visits, v0 (pre-infection) for the baseline, and 4 wks (v9) and 24 wks (v14) post-infection or purified antibodies, CH58 and CH65 were added to the respective wells. After 1 hr incubation at 37 °C and 5% $CO_2$, the plate was centrifuged and washed with wash buffer. After washing, the cells were resuspended in wash buffer and the plate was read using the BD LSRII or BD LSRFortessa with the High Throughput Sampler (HTS) with a minimum of 1250 events, to detect the activity of granzyme B (GzB) released by the effector population into target cells. The viable target cells with activated GzB substrate represented the actual population recognized by the effector cells and reported as %GzB activity. The results are reported after background subtraction of the signal acquired from target cells incubated with effector cells in the absence of plasma/antibodies.

## Mouse immunizations

Six-week-old female BALB/c (Strain: 000651) mice were received from Jackson's laboratory for immunization experiments. The immunization was initiated after 2 wks of quarantine. The weights of the mice were taken periodically from the start of the quarantine period to assess growth and health of mice throughout the experiment. Twenty μg of the antigen complexed with Alhydrogel 2% (Invivogen) as adjuvant was injected intramuscularly per mouse using a 22–23 gauze needle syringe. Three boosters were given after prime/first immunization at an interval of 3 wks and tail bleeds were performed to collect sera before each immunization. Mice were also bled before the first immunization to collect pre-immunized (pre-bleed) sera for the negative control. Terminal bleed was performed through cardiac puncture under general anesthesia followed by cervical dislocation to euthanize the animals.

## Statistical analyses

Statistical information is provided in the figure legends. Statistical analysis was performed using GraphPad Prism 8 and unpaired Student's $t$-test was applied to calculate significance where mentioned. Data were considered statistically significant at *$p \leq 0.05$, **$p \leq 0.01$, ***$p \leq 0.001$ and ****$p \leq 0.0001$. SPR binding data of the vaccinated mice sera was analyzed through paired non-parametric 1-way ANOVA (Friedman test) to calculate statistical significance between different peptides in the same group with a confidence interval of 0.05.

## Acknowledgements

We thank the study participants who volunteered in the RV217 study and provided their samples which have supported numerous follow-up studies for HIV vaccine research including our study reported here. This work was supported by the National Institute of Allergy and Infectious Diseases NIH grants AI111538 and AI102725 to V.B.R. and by a cooperative agreement (W81XWH-07-2-0067 and W81XWH-11–0174) between the Henry M Jackson Foundation for the Advancement of Military Medicine, Inc and the U.S. Department of Defense. DK acknowledges supports by the National Institutes of Health (R01GM133840 and R01GM123055) and the National Science Foundation (CMMI1825941, MCB1925643, IIS2211598, DMS2151678, DBI2146026, and DBI2003635).

## Additional information

### Competing interests

Gustavo Kijak: Gustavo Kijak is affiliated with AstraZeneca and owns stocks/shares. The author has no other competing interests to declare. Venigalla B Rao: The other authors declare that no competing interests exist.

## Funding

| Funder | Grant reference number | Author |
| --- | --- | --- |
| National Institute of Allergy and Infectious Diseases | AI111538 | Venigalla B Rao |
| Walter Reed Army Institute of Research | W81XWH-07-2-0067 | Venigalla B Rao |
| National Institutes of Health | R01GM133840 | Daisuke Kihara |
| National Science Foundation | CMMI1825941 | Daisuke Kihara |
| National Institute of Allergy and Infectious Diseases | AI102725 | Venigalla B Rao |
| Walter Reed Army Institute of Research | W81XWH-11-0174 | Venigalla B Rao |
| National Institutes of Health | R01GM123055 | Daisuke Kihara |
| National Science Foundation | MCB1925643 | Daisuke Kihara |
| National Science Foundation | IIS2211598 | Daisuke Kihara |
| National Science Foundation | DMS2151678 | Daisuke Kihara |
| National Science Foundation | DBI2146026 | Daisuke Kihara |
| National Science Foundation | DBI2003635 | Daisuke Kihara |

The funders had no role in study design, data collection and interpretation, or the decision to submit the work for publication.

## Author contributions

Swati Jain, Conceptualization, Investigation, Visualization, Methodology, Writing – original draft, Writing – review and editing; Gherman Uritskiy, Marthandan Mahalingam, Himanshu Batra, Subhash Chand, Hung V Trinh, Charles Beck, Woong-Hee Shin, Wadad Alsalmi, Gustavo Kijak, Leigh A Eller, Jerome Kim, Sodsai Tovanabutra, Investigation; Daisuke Kihara, Guido Ferrari, Merlin L Robb, Mangala Rao, Investigation, Writing – review and editing; Venigalla B Rao, Conceptualization, Investigation, Methodology, Visualization, Writing – original draft, Writing – review and editing

## Author ORCIDs

Swati Jain ![ORCID] http://orcid.org/0000-0002-8625-8218
Venigalla B Rao ![ORCID] https://orcid.org/0000-0002-0777-6587

## Ethics

Mice used in the study were maintained in the pathogen-free animal facility at the Catholic University of America, Washington, D.C. All animal protocols conducted for the current study were reviewed and approved by the Institutional Animal Care and Use Committee (IACUC) at the Catholic University of America under Animal Assurance Number- A4431-01.

Reviewer #1 (Public Review): https://doi.org/10.7554/eLife.92379.3.sa1
Reviewer #2 (Public Review): https://doi.org/10.7554/eLife.92379.3.sa2
Author response https://doi.org/10.7554/eLife.92379.3.sa3

## Additional files

### Supplementary files
• MDAR checklist

### Data availability

All data generated or analysed during this study are included in the manuscript and supporting files; source data files have been provided for Figure 2-6 and 8 which include uncropped full western blot images and data used to generate graphs. Virus sequences used in the study were previously deposited in GenBank under the following accession numbers: Participant 7: MN791995 – MN792033, MK656525 – MK656553, Participant 61: KY580541 – KY580724, Participant 94: MN792035 – MN792075, Participant 100: MN792076 – MN792116, KY580582 – KY580642.

The following previously published datasets were used:

| Author(s) | Year | Dataset title | Dataset URL | Database and Identifier |
|---|---|---|---|---|
| Rolland et al | 2019 | HIV-1 isolate 40007v02_01R from Thailand pol protein (pol) gene, partial cds; and vif protein (vif), vpr protein (vpr), tat protein (tat), rev protein (rev), vpu protein (vpu), envelope glycoprotein (env), and nef protein (nef) genes, complete cds | https://www.ncbi.nlm.nih.gov/nuccore/MN791995 | NCBI Nucleotide, MN791995 |
| Rolland et al | 2019 | HIV-1 isolate 40007v14_11 from Thailand, partial genome | https://www.ncbi.nlm.nih.gov/nuccore/MN792033 | NCBI Nucleotide, MN792033 |
| Trinh et al | 2019 | HIV-1 isolate 40007v02_01R from Thailand envelope glycoprotein (env) and vpu protein (vpu) genes, partial cds | https://www.ncbi.nlm.nih.gov/nuccore/MK656525 | NCBI Nucleotide, MK656525 |
| Trinh et al | 2019 | HIV-1 isolate 40007v14_11 from Thailand envelope glycoprotein (env) and vpu protein (vpu) genes, partial cds | https://www.ncbi.nlm.nih.gov/nuccore/MK656553 | NCBI Nucleotide, MK656553 |
| Kijak et al | 2017 | HIV-1 isolate 40061v03_01 from Thailand gag protein (gag) gene, complete cds; pol protein (pol) gene, partial cds; and vif protein (vif), vpr protein (vpr), tat protein (tat), rev protein (rev), vpu protein (vpu), envelope glycoprotein (env), and nef protein (nef) genes, complete cds | https://www.ncbi.nlm.nih.gov/nuccore/KY580541 | NCBI Nucleotide, KY580541 |
| Kijak et al | 2017 | HIV-1 isolate 40061v14_11R from Thailand nonfunctional pol protein (pol) gene, partial sequence; and vif protein (vif), vpr protein (vpr), tat protein (tat), rev protein (rev), vpu protein (vpu), envelope glycoprotein (env), and nef protein (nef) genes, complete cds | https://www.ncbi.nlm.nih.gov/nuccore/KY580724 | NCBI Nucleotide, KY580724 |

*Continued on next page*

*Continued*

| Author(s) | Year | Dataset title | Dataset URL | Database and Identifier |
|---|---|---|---|---|
| Rolland et al | 2019 | HIV-1 isolate 40094v01_01R from Thailand pol protein (pol) gene, partial cds; and vif protein (vif), vpr protein (vpr), tat protein (tat), rev protein (rev), vpu protein (vpu), envelope glycoprotein (env), and nef protein (nef) genes, complete cds | https://www.ncbi.nlm.nih.gov/nuccore/MN792035 | NCBI Nucleotide, MN792035 |
| Rolland et al | 2019 | HIV-1 isolate 40094v14_11 from Thailand, partial genome | https://www.ncbi.nlm.nih.gov/nuccore/MN792075 | NCBI Nucleotide, MN792075 |
| Rolland et al | 2019 | HIV-1 isolate 40100v01_env01 from Thailand rev protein (rev) and tat protein (tat) genes, partial cds; vpu protein (vpu) and envelope glycoprotein (env) genes, complete cds; and nef protein (nef) gene, partial cds | https://www.ncbi.nlm.nih.gov/nuccore/MN792076 | NCBI Nucleotide, MN792076 |
| Rolland et al | 2019 | HIV-1 isolate 40100v14_07 from Thailand, partial genome | https://www.ncbi.nlm.nih.gov/nuccore/MN792116 | NCBI Nucleotide, MN792116 |
| Kijak et al | 2017 | HIV-1 isolate 40100v01_01 from Thailand gag protein (gag) gene, complete cds; pol protein (pol) gene, partial cds; and vif protein (vif), vpr protein (vpr), tat protein (tat), rev protein (rev), vpu protein (vpu), envelope glycoprotein (env), and nef protein (nef) genes, complete cds | https://www.ncbi.nlm.nih.gov/nuccore/KY580582 | NCBI Nucleotide, KY580582 |
| Kijak et al | 2017 | HIV-1 isolate 40100v14_11 from Thailand gag protein (gag) gene, complete cds; pol protein (pol) gene, partial cds; and vif protein (vif), vpr protein (vpr), tat protein (tat), rev protein (rev), vpu protein (vpu), envelope glycoprotein (env), and nef protein (nef) genes, complete cds | https://www.ncbi.nlm.nih.gov/nuccore/KY580642 | NCBI Nucleotide, KY580642 |

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
