## [Editor Report · eLife assessment]

This study provides a detailed evaluation of how HIV evades nascent immune pressure from people living with HIV followed nearly immediately after infection. There is **convincing** evidence that H173 mutations in the V2 loop was a key determinant of selection pressure and escape. These data are congruent with protection in the RV144 clinical trial, the only trial that showed protection from infection. Overall, this study is an **important** contribution to the field.

---

## [Referee Report · Reviewer #1 (Public Review)]

Summary:

This study used a unique acute HIV-1 infection cohort, RV217, to study the evolution of transmitted founder viral Envelope sequences under nascent immune pressure. The striking feature of the RV217 cohort is the ability to detect viremia in the first week of infection, which can be followed at discrete Fiebig stages over long time intervals. This study evaluated Env sequences at 1 week, 4 weeks, and 24 weeks to provide a picture of viral and immunological co-evolution from Fiebig Stage I (1 week), Fiebig Stages IV (4 weeks), and Fiebig Stage VI (>24 weeks). This study design enabled lineage tracing of viral variants from a single transmitted founder (T/F) over the Fiebig Stages I, IV, and VI under nascent immune pressure generated in response to the T/F virus and its subsequent mutants.

Strengths:

As expected, there were temporal differences in the appearance of virus quasispecies among the individuals, which were located predominantly in solvent-exposed residues of Env, which is consistent with prior literature. Interestingly, two waves of antibody reactivity were observed for variants with mutations in the V2 region that harbors V2i and V2p epitopes correlated with protection in the RV144 clinical trial. Two waves of antibody response, detected by SPR, were observed, with the first wave being predominated by antibodies specific for the T/F07 V2 epitope associated with H173 located on the C β-strand in the V2 region. The second wave was dominated by antibodies specific for an H to Y mutation at 173 that emerged due to antibody-mediated pressure to the original H173 virus. This is a remarkable finding in three ways.

First, the mutation is in the C β-strand, an unlikely paratope contact residue, as this region of the V2 loop is shielded by glycans in Env trimer structures with full glycan representation (see PDB:5t3x). The structure used for modeling in the current study was an earlier structure, PDB:4TVP, that had many truncated glycans. This does not detract from the finding that the H173Y mutation likely causes a conformational shift from a more rigid helical/coil conformation to a more dynamic conformation with a β-stranded and β-sheet core preference as indicated by the literature and by the MD simulations carried out by the authors. This observation suggests that using V2 scaffolds with both the H173 and H173Y variants will increase the breadth of potentially protective antibody responses to HIV-1, as indicated in reference 42, cited by the authors. Interestingly, the H173Y mutation abrogates reactivity to mAb CH58 and attenuates reactivity to mAb CH59 isolated from RV144 volunteers. These mAbs recognize conformationally distinct V2 epitopes, adding further credence to the conclusion that the H173Y mutation results in a conformational switch of the V2 region.

Second, the H173Y mutation affects the conformation of V2 epitopes recognized by mAbs that do not neutralize T/F HIV-1 while mediating potent ADCC. The ADCC data in the manuscript provides strong support for this hypothesis and augers for further exploration of the V2 epitopes as HIV-1 vaccine targets.

Third, it is striking that immunogens based on the H173Y mutation successfully recapitulated the observed human antibody responses in wild-type Balb/c mice. The investigators used their extensive knowledge of V2 structures and scaffold-immunogens to create the library of constructs used for this study. In this case, the ΔDSV mutation increased the breadth and magnitude of the murine antibody responses.

---

## [Referee Report · Reviewer #2 (Public Review)]

Summary:

In this study, researchers aimed to understand how a transmitted/founder (T/F) HIV virus escapes host immune pressure during early infection. They focused on the V1V2 domain of the HIV-1 envelope protein, a key determinant of virus escape. The study involved four participants from the RV217 Early Capture HIV Cohort (ECHO) project, which allowed tracking HIV infection from just days after infection.

The study identified a significant H173Y escape mutation in the V2 domain of a T/F virus from one participant. This mutation, located in the relatively conserved "C" β-strand, was linked to viral escape against host immune pressure. The study further investigated the epitope specificity of antibodies in the participant's plasma, revealing that the H173Y mutation played a crucial role in epitope switching during virus escape. Monoclonal antibodies from the RV144 vaccine trial, CH58, and CH59, showed reduced binding to the V1V2-Y173 escape variant. Additionally, the study examined antibody-dependent cellular cytotoxicity (ADCC) responses and found resistance to killing in the Y173 mutants. The H173Y mutation was identified as the key variant selected against the host's immune pressure directed at the V2 domain.

The researchers hypothesized that the H173Y mutation caused a structural/conformational change in the C β-strand epitope, leading to viral escape. This was supported by molecular dynamics simulations and structural modeling analyses. They then designed combinatorial V2 immunogen libraries based on natural HIV-1 sequence diversity, aiming to broaden antibody responses. Mouse immunizations with these libraries demonstrated enhanced recognition of diverse Env antigens, suggesting a potential strategy for developing a more effective HIV vaccine.

In summary, the study provides insights into the early evolution of HIV-1 during infection, highlighting the importance of the V1V2 domain and identifying key escape mutations. The findings suggest a novel approach for designing HIV vaccine candidates that consider the diversity of escape mutations to induce broader and more effective immune responses.

Strengths:

The article presents several strengths:

(1) The experimental design is well-structured, involving multiple stages from phylogenetic analyses to mouse model testing, providing a comprehensive approach to studying virus escape mutations.

(2) The study utilizes a unique dataset from the RV217 Early Capture HIV Cohort (ECHO) project, allowing for the tracking of HIV infection from the very early stages in the absence of antiretroviral therapy. This provides valuable insights into the evolution of the virus.

(3) The use of advanced techniques such as phylogenetic analyses, nanoscaffold technology, controlled mutagenesis, and monoclonal antibody evaluations demonstrates the application of cutting-edge methodologies in the study.

(4) The research goes beyond genetic analysis and provides an in-depth characterization of the escape mutation's impact, including structural analyses through Molecular Dynamics simulations, antibody responses, and functional implications for virus survival.

(5) The study provides insights into the immune responses triggered by the escape mutation, including the specificity of antibodies and their ability to recognize diverse HIV-1 Env antigens.

(7) The exploration of combinatorial immunogen libraries is a strength, as it offers a novel approach to broaden antibody responses, providing a potential avenue for future vaccine design.

(8) The research is highly relevant to vaccine development, as it sheds light on the dynamics of HIV escape mutations and their interaction with the host immune system. This information is crucial for designing effective vaccines that can preemptively interfere with viral acquisition.

(9) The study integrates findings from virology, immunology, structural biology, and bioinformatics, showcasing an interdisciplinary approach that enhances the depth and breadth of the research.

(10) The article is well-written, with a clear presentation of methods, results, and implications, making it accessible to both specialists and a broader scientific audience.

---

## [Author Response]

The following is the authors’ response to the original reviews.

**Reviewer #1 (Recommendations For The Authors):**
(1) V2 epitopes exhibit properties of CD4i epitopes in that they are largely absent from the native Env surface, probably by glycan-occlusion, but become more exposed upon CD4 binding. Although the V2-scaffolds were produced in GnTi- cells to produce highmannose proteins, it appears that no systematic analysis of glycan content or structure was carried out save for enzymatic deglycosylation of the constructs to sharpen bands on SDS-PAGE gels. It would be helpful if the authors could comment on how the lack of this information might impact their conclusions.

We thank the reviewer for this comment.

The lack of native glycan structures is a common phenomenon in all HIV studies involving in vitro cell culture-expressed envelope proteins.

As the reviewer mentioned, it is clear that our V1V2 scaffolds produced in GnTi-cells contain the expected high-mannose glycans, as evident from a significant shift and sharpening of the protein bands on the SDS-PAGE gel upon deglycosylation with the PNGase enzyme.

In our previously published studies by Chand et al.,2017* (ref. below), the V1V2 scaffolds were shown to bind to glycan-dependent PG9 antibody suggesting that the conformation of the PG9 epitope is retained in the high-mannose V1V2 scaffold. This information has also been added to the “Hypothesis and Experimental Design” section of the Results in the revised manuscript.

Additionally, as shown in Results, the human antibodies elicited in study participants against native glycosylated envelope protein due to natural HIV-1 infection distinguished the H173 and Y 173 epitopes in the high-mannose scaffolds, which was also recapitulated in our mouse studies using the GnTi-expressed high-mannose V1V2 scaffolds as antigens.

Therefore, it does not seem likely that differences in glycans per se majorly affected the binding or the conclusions from our studies.

*Chand S, Messina EL, AlSalmi W, Ananthaswamy N, Gao G, Uritskiy G, Padilla-Sanchez V, Mahalingam M, Peachman KK, Robb ML, Rao M, Rao VB. Glycosylation and oligomeric state of the envelope protein might influence HIV-1 virion capture by α4β7 integrin. Virology. 2017 Aug;508:199-212. doi:10.1016/j.virol.2017.05.016. Epub 2017 May 31. PMID: 28577856; PMCID: PMC5526109.

(2) Similarly, the MD simulations appear to be performed without taking glycan structure/occupancy.

We were unable to perform glycan-dependent MD simulation studies because of the high computational demands and also the technical limitations that existed at the time of the study several years ago. Therefore, we focused on the protein backbone of the short C-strand in the V2 region that lacks glycan sites and in previous published studies has been demonstrated as conformationally polymorphic.

Since this C-strand epitope is the binding site for many V2-directed antibodies identified previously, we hypothesized that it is relevant to explore this small immunogenic epitope for its propensity to change conformation due to an escape mutation discovered at residue 173 in a natural HIV-1 infection. How might this epitope behave in MD simulations in the presence of different glycans requires further investigation.